# Learning to Relax: Setting Solver Parameters Across a Sequence of Linear System Instances

**Mikhail Khodak**
CMU
khodak@cmu.edu

**Edmond Chow**
Georgia Tech.
echow@cc.gatech.edu

**Maria-Florina Balcan**
CMU
ninamf@cs.cmu.edu

**Ameet Talwalkar**
CMU
talwalkar@cmu.edu

## Abstract

Solving a linear system $\mathbf{Ax} = \mathbf{b}$ is a fundamental scientific computing primitive for which numerous solvers and preconditioners have been developed. These come with parameters whose optimal values depend on the system being solved and are often impossible or too expensive to identify; thus in practice sub-optimal heuristics are used. We consider the common setting in which many related linear systems need to be solved, e.g. during a single numerical simulation. In this scenario, can we sequentially choose parameters that attain a near-optimal overall number of iterations, without extra matrix computations? We answer in the affirmative for Successive Over-Relaxation (SOR), a standard solver whose parameter $\omega$ has a strong impact on its runtime. For this method, we prove that a bandit online learning algorithm—using only the number of iterations as feedback—can select parameters for a sequence of instances such that the overall cost approaches that of the best fixed $\omega$ as the sequence length increases. Furthermore, when given additional structural information, we show that a *contextual* bandit method asymptotically achieves the performance of the *instance-optimal* policy, which selects the best $\omega$ for each instance. Our work provides the first learning-theoretic treatment of high-precision linear system solvers and the first end-to-end guarantees for data-driven scientific computing, demonstrating theoretically the potential to speed up numerical methods using well-understood learning algorithms.

## 1 Introduction

The bottleneck subroutine in many scientific computations is a solver returning an approximate solution to a linear system. For example, simulating a partial differential equation (PDE) often involves solving sequences of high-dimensional systems to very high precision (Thomas, 1999). A vast array of solvers and preconditioners have thus been developed, many of which have tunable parameters that significantly affect runtime (Greenbaum, 1997; Hackbusch, 2016). There is a long literature analyzing these algorithms, and indeed for some problems we have a strong understanding of the optimal parameters for a given matrix. However, computing them can be more costly than solving the original system, leading to an assortment of heuristics for setting good parameters (Ehrlich, 1981; Golub & Ye, 1999).

We provide an alternative to such heuristics by taking advantage of the fact that we often sequentially solve *many* linear systems. In addition to numerical simulation, this occurs in graphics computations such as mean-curvature flow (Kazhdan et al., 2012), nonlinear system solvers (Marquardt, 1963), and beyond. A natural approach is to treat these instances as data to be passed to a machine learning (ML) algorithm; in particular the framework of online learning (Cesa-Bianchi & Lugosi, 2006) provides a language to reason about such sequential learning problems. For example, if we otherwise would solve a sequence of linear systems $(\mathbf{A}_1, \mathbf{b}_1), \ldots, (\mathbf{A}_T, \mathbf{b}_T)$ using a given solver with a fixed parameter, can we use ML to do as well as the best choice of that parameter, i.e. can we *minimize regret*? Or, if the matrices are all diagonal shifts of single matrix $\mathbf{A}$, can we learn the functional relationship between the shift $c_t$ and the optimal solver parameter for $\mathbf{A}_t = \mathbf{A} + c_t \mathbf{I}_n$, i.e. can we predict using *context*?

We investigate these questions for the Successive Over-Relaxation (SOR) solver, a generalization of Gauss-Seidel whose relaxation parameter $\omega \in (0, 2)$ dramatically affects the number of iterations (c.f. Figure 1, noting the log-scale). SOR and its symmetric variant are well-studied and often used as preconditioners for Krylov methods such as conjugate gradient (CG), as bases for semi-iterative schemes, and as multigrid smoothers. Analogous to some past setups in data-driven algorithms (Balcan et al., 2018; Khodak et al., 2022), we sequentially set the parameter $\omega_t$ for SOR to use when solving each lin-

ear system $(\mathbf{A}_t, \mathbf{b}_t)$. Unlike past theoretical studies of related methods (Gupta & Roughgarden, 2017; Bartlett et al., 2022; Balcan et al., 2022), we aim to provide *end-to-end* guarantees—covering the full pipeline from data-intake to efficient learning to execution—while minimizing dependence on the dimension ($n$ can be $10^5$ or higher) and precision ($1/\varepsilon$ can be $10^8$ or higher). We emphasize that we do *not* seek to immediately improve the empirical state of the art, and also that existing research on saving computation when solving sequences of linear systems (recycling Krylov subspaces, reusing preconditioners, etc.) is complementary to our own, i.e. it can be used in addition to the ideas presented here.

## 1.1 Core contributions

We study two distinct theoretical settings, corresponding to views on the problem from two different approaches to data-driven algorithms. In the first we have a deterministic sequence of instances and study the spectral radius of the iteration matrix, the main quantity of interest in classical analysis of SOR (Young, 1971). We show how to convert its asymptotic guarantee into a surrogate loss that upper bounds the number of iterations via a quality measure of the chosen parameter, in the style of *algorithms with predictions* (Mitzenmacher & Vassilvitskii, 2021). The bound holds under a *near-asymptotic* condition implying that convergence occurs near the asymptotic regime, i.e. when the spectral radius of the iteration matrix governs the convergence. We verify the assumption and show that one can learn the surrogate losses using only bandit feedback from the original costs; notably, despite being non-Lipschitz, we take advantage of the losses' unimodal structure to match the optimal $\tilde{\mathcal{O}}(T^{2/3})$ regret for Lipschitz bandits (Kleinberg, 2004). Our bound also depends only logarithmically on the precision and not at all on the dimension. Furthermore, we extend to the diagonally shifted setting described before, showing that an efficient, albeit pessimistic, contextual bandit (CB) method has $\tilde{\mathcal{O}}(T^{3/4})$ regret w.r.t. the instance-optimal policy that always picks the best $\omega_t$. Finally, we show a similar analysis of learning a relaxation parameter for the more popular (symmetric SOR-preconditioned) CG method.

Our second setting is *semi-stochastic*, with target vectors $\mathbf{b}_t$ drawn i.i.d. from a (radially truncated) Gaussian. This is a reasonable simplification, as convergence usually depends more strongly on $\mathbf{A}_t$, on which we make no extra assumptions. We show that the expected cost of running a symmetric variant of SOR (SSOR) is $\mathcal{O}(\sqrt{n})\text{polylog}(\frac{n}{\varepsilon})$-Lipschitz w.r.t. $\omega$, so we can (a) compete with the optimal number of iterations—rather than with the best upper bound—and (b) analyze more practical, regression-based CB algorithms (Foster & Rakhlin, 2020; Simchi-Levi & Xu, 2021). We then show $\tilde{\mathcal{O}}(\sqrt[3]{T^2\sqrt{n}})$ regret when comparing to the single best $\omega$ and $\tilde{\mathcal{O}}(T^{9/11}\sqrt{n})$ regret w.r.t. the instance-optimal policy in the diagonally shifted setting using a novel, Chebyshev regression-based CB algorithm. While the results do depend on the dimension $n$, the dependence is much weaker than that of past work on data-driven tuning of a related regression problem (Balcan et al., 2022).

**Remark 1.1.** *Likely the most popular algorithms for linear systems are Krylov subspace methods such as CG. While an eventual aim of our line of work is to understand how to tune (many) parameters of (preconditioned) CG and other algorithms, SOR is a well-studied method and serves as a meaningful starting point. In fact, we show that our near-asymptotic analysis extends directly, and in the semi-stochastic setting there is a natural path to (e.g.) SSOR-preconditioned CG, as it can be viewed as computing polynomials of iteration matrices where SSOR just takes powers. Lastly, apart from its use as a preconditioner and smoother, SOR is still sometimes preferred for direct use as well (Fried & Metzler, 1978; Van Vleck & Dwyer, 1985; King et al., 1987; Woźnicki, 1993; 2001).*

## 1.2 Technical and theoretical contributions

By studying a scientific computing problem through the lens of data-driven algorithms and online learning, we also make the following contributions to the latter two fields:

1. Ours is the first head-to-head comparison of two leading theoretical approaches to data-driven algorithms applied to the same problem. While the algorithms with predictions approach in Section 2 takes better advantage of the scientific computing literature to obtain (arguably) more interpretable and dimension-independent bounds, data-driven algorithm design (Balcan, 2021) competes directly with the quantity of interest in Section 3 and enables guarantees for modern CB algorithms.
2. For algorithms with predictions, our near-asymptotic approach may be extendable to other iterative solvers, as we demonstrate with CG. We also show that such performance bounds on a (partially-observable) cost are learnable even when the bounds themselves are too expensive to compute.
3. In data-driven algorithm design, we take the novel theoretical approach of proving continuity of *the expectation of* a discrete cost, rather than showing dispersion of its discontinuities (Balcan et al., 2018) or bounding predicate complexity (Bartlett et al., 2022).

---

**Algorithm 1:** Successive over-relaxation (SOR) with a relative convergence condition.

**Input:** $\mathbf{A} \in \mathbb{R}^{n \times n}$, $\mathbf{b} \in \mathbb{R}^n$, parameter $\omega \in (0, 2)$, initial vector $\mathbf{x} \in \mathbb{R}^n$, tolerance $\varepsilon > 0$

$\mathbf{D} + \mathbf{L} + \mathbf{L}^T \leftarrow \mathbf{A}$   // $\mathbf{D}$ is diagonal, $\mathbf{L}$ is strictly lower triangular

$\mathbf{W}_\omega \leftarrow \mathbf{D}/\omega + \mathbf{L}$              // compute the third normal form

$\mathbf{r}_0 \leftarrow \mathbf{b} - \mathbf{A}\mathbf{x}$                   // compute initial residual

**for** $k = 0, \dots$ **do**

    **if** $\|\mathbf{r}_k\|_2 \leq \varepsilon \|\mathbf{r}_0\|_2$ **then**

        $\lfloor$ **return** $k$     // return iteration count (for use in learning)

    $\mathbf{x} = \mathbf{x} + \mathbf{W}_\omega^{-1}\mathbf{r}_k$  // update vector after solving triangular system

    $\mathbf{r}_{k+1} \leftarrow \mathbf{b} - \mathbf{A}\mathbf{x}$              // compute the next residual

---

4. We introduce the idea of using CB to set *instance-adaptive* algorithmic parameters; while (linear) instance-adaptivity was also shown via convexity by Khodak et al. (2022), we go further by taking advantage of multi-instance structure to asymptotically do as well as the *instance-optimal* policy.
5. We show that standard discretization-based bandit algorithms are optimal for sequences of adversarially chosen *semi-Lipschitz* losses that generalize regular Lipschitz functions (c.f. Appendix B).
6. We introduce a new CB method that combines SquareCB (Foster & Rakhlin, 2020) with Chebyshev polynomial regression to get sublinear regret on Lipschitz losses (c.f. Appendix C).

### 1.3 RELATED WORK AND COMPARISONS

We discuss the existing literature on solving sequences of linear systems (Parks et al., 2006; Tebbens & Tůma, 2007; Elbouyahyaoui et al., 2021), work integrating ML with scientific computing to amortize cost (Amos, 2023; Arisaka & Li, 2023), and past theoretical studies of data-driven algorithms (Gupta & Roughgarden, 2017; Balcan et al., 2022) in Appendix A. For the latter we include a detailed comparison of the generalization implications of our work with the GJ framework (Bartlett et al., 2022). Lastly, we address the baseline of approximating the spectral radius of the Jacobi iteration matrix.

## 2 ASYMPTOTIC ANALYSIS OF LEARNING THE RELAXATION PARAMETER

We start this section by going over the problem setup and the SOR solver. Then we consider the asymptotic analysis of the method to derive a reasonable performance upper bound to target as a surrogate loss for the true cost function. Finally, we prove and analyze online learning guarantees.

### 2.1 SETUP

At each step $t = 1, \dots, T$ of (say) a numerical simulation we get a linear system instance, defined by a matrix-vector pair $(\mathbf{A}_t, \mathbf{b}_t) \in \mathbb{R}^{n \times n} \times \mathbb{R}^n$, and are asked for a vector $\mathbf{x} \in \mathbb{R}^n$ such that the norm of its *residual* or *defect* $\mathbf{r} = \mathbf{b}_t - \mathbf{A}_t\mathbf{x}$ is small. For now we define "small" in a relative sense, specifically $\|\mathbf{A}_t\mathbf{x} - \mathbf{b}_t\|_2 \leq \varepsilon\|\mathbf{b}_t\|_2$ for some *tolerance* $\varepsilon \in (0, 1)$; note that when using an iterative method initialized at $\mathbf{x} = \mathbf{0}_n$ this corresponds to reducing the residual by a factor $1/\varepsilon$, which we call the *precision*. In applications it can be quite high, and so we will show results whose dependence on it is at worst logarithmic. To make the analysis tractable, we make two assumptions (for now) about the matrices $\mathbf{A}$: they are symmetric positive-definite and consistently-ordered (c.f. Hackbusch (2016, Definition 4.23)). We emphasize that, while not necessary for convergence, both are standard in the analysis of SOR (Young, 1971); see Hackbusch (2016, Criterion 4.24) for multiple settings where they holds.

To find a suitable $\mathbf{x}$ for each instance in the sequence we apply Algorithm 1 (SOR), which at a high-level works by multiplying the current residual $\mathbf{r}$ by the inverse of a matrix $\mathbf{W}_\omega$—derived from the diagonal $\mathbf{D}$ and lower-triangular component $\mathbf{L}$ of $\mathbf{A}$—and then adding the result to the current iterate $\mathbf{x}$. Note that multiplication by $\mathbf{W}_\omega^{-1}$ is efficient because $\mathbf{W}_\omega$ is triangular. We will measure the cost of this algorithm by the number of iterations it takes to reach convergence, which we denote by $\texttt{SOR}(\mathbf{A}, \mathbf{b}, \omega)$, or $\texttt{SOR}_t(\omega)$ for short when it is run on the instance $(\mathbf{A}_t, \mathbf{b}_t)$. For simplicity, we will assume that the algorithm is always initialized at $\mathbf{x} = \mathbf{0}_n$, and so the first residual is just $\mathbf{b}$.

Having specified the computational setting, we now turn to the learning objective, which is to sequentially set the parameters $\omega_1, \dots, \omega_T$ so as to minimize the total number of iterations:

$$\sum\nolimits_{t=1}^{T} \texttt{SOR}_t(\omega_t) = \sum\nolimits_{t=1}^{T} \texttt{SOR}(\mathbf{A}_t, \mathbf{b}_t, \omega_t) \tag{1}$$

To set $\omega_t$ at some time $t > 1$, we allow the learning algorithm access to the costs $\text{SOR}_s(\omega_s)$ incurred at the previous steps $s = 1, \ldots, t - 1$; in the literature on online learning this is referred to as the *bandit* or *partial feedback* setting, to distinguish from the (easier, but unreasonable for us) *full information* case where we have access to the cost function $\text{SOR}_s$ at every $\omega$ in its domain.

Selecting the optimal $\omega_t$ using no information about $\mathbf{A}_t$ is impossible, so we must use a *comparator* to obtain an achievable measure of performance. In online learning this is done by comparing the total cost incurred (1) to the counterfactual cost had we used a *single*, best-in-hindsight $\omega$ at every timestep $t$. We take the minimum over some domain $\Omega \subset (0, 2)$, as SOR diverges outside it. While in some settings we will compete with every $\omega \in (0, 2)$, we will often algorithmically use $[1, \omega_{\max}]$ for some $\omega_{\max} < 2$. The upper limit ensures a bound on the number of iterations—required by bandit algorithms—and the lower limit excludes $\omega < 1$, which is rarely used because theoretical convergence of vanilla SOR is worse there for realistic problems, e.g. those satisfying our assumptions.

This comparison-based approach for measuring performance is standard in online learning and effectively assumes a good $\omega \in \Omega$ that does well-enough on all problems; in Figure 1 (center-left) we show that this is sometimes the case. However, the center-right plot in the same figure shows we might do better by using additional knowledge about the instance; in online learning this is termed a *context* and there has been extensive development of contextual bandit algorithms that do as well as the best fixed policy mapping contexts to predictions. We will study an example of this in the *diagonally shifted* setting, in which $\mathbf{A}_t = \mathbf{A} + c_t \mathbf{I}_n$ for scalars $c_t \in \mathbb{R}$; while mathematically simple, this structure arises in natural settings, e.g. solving the heat equation with temporally variable diffusivity, and is well-motivated by other applications (Frommer & Glässner, 1998; Bellavia et al., 2011; Baumann & van Gijzen, 2015; Anzt et al., 2016; Wang et al., 2019). Furthermore, the same learning algorithms can also be extended to make use of other context information, e.g. rough spectral estimates.

## 2.2 ESTABLISHING A SURROGATE UPPER BOUND

Our first goal is to solve $T$ linear systems almost as fast as if we had used the best fixed $\omega \in \Omega$. In online learning, this corresponds to minimizing *regret*, which for cost functions $\ell_t : \Omega \mapsto \mathbb{R}$ is defined as

$$\text{Regret}_\Omega(\{\ell_t\}_{t=1}^T) = \sum_{t=1}^T \ell_t(\omega_t) - \min_{\omega \in \Omega} \sum_{t=1}^T \ell_t(\omega) \tag{2}$$

In particular, since we can upper-bound the objective (1) by $\text{Regret}_\Omega(\{\text{SOR}_t\}_{t=1}^T)$ plus the optimal cost $\min_{\omega \in \Omega} \sum_{t=1}^T \text{SOR}_t(\omega)$, if we show that regret is *sublinear* in $T$ then the leading-order term in the upper bound corresponds to the cost incurred by the optimal fixed $\omega$.

Many algorithms attaining sublinear regret under different conditions on the losses $\ell_t$ have been developed (Cesa-Bianchi & Lugosi, 2006; Bubeck & Cesa-Bianchi, 2012). However, few handle losses with discontinuities—i.e. most algorithmic costs—and those that do (necessarily) need additional conditions on their locations (Balcan et al., 2018; 2020). At the same time, numerical analysis often deals more directly with continuous asymptotic surrogates for cost, such as convergence rates. Taking inspiration from this, and from the algorithms with predictions idea of deriving surrogate loss functions for algorithmic costs (Khodak et al., 2022), in this section we instead focus on finding *upper bounds* $U_t$ on $\text{SOR}_t$ that are both (a) learnable and (b) reasonably tight in-practice. We can then aim for overall performance nearly as good as the optimal $\omega \in \Omega$ as measured by these upper bounds:

$$\sum_{t=1}^T \text{SOR}_t(\omega_t) \leq \sum_{t=1}^T U_t(\omega_t) = \text{Regret}_\Omega(\{U_t\}_{t=1}^T) + \min_{\omega \in \Omega} \sum_{t=1}^T U_t(\omega) = o(T) + \min_{\omega \in \Omega} \sum_{t=1}^T U_t(\omega) \tag{3}$$

A natural approach to get a bound $U_t$ is via the *defect reduction matrix* $\mathbf{C}_\omega = \mathbf{I}_n - \mathbf{A}(\mathbf{D}/\omega + \mathbf{L})^{-1}$, so named because the residual at iteration $k$ is equal to $\mathbf{C}_\omega^k \mathbf{b}$ and $\mathbf{b}$ is the first residual. Under our assumptions on $\mathbf{A}$, Young (1971) shows that the spectral radius $\rho(\mathbf{C}_\omega)$ of $\mathbf{C}_\omega$ is a (nontrivial to compute) piecewise function of $\omega$ with a unique minimum in $[1, 2)$. Since we have error $\|\mathbf{C}_\omega^k \mathbf{b}\|_2 / \|\mathbf{b}\|_2 \leq \|\mathbf{C}_\omega^k\|_2$ at iteration $k$, $\rho(\mathbf{C}_\omega) = \lim_{k \to \infty} \sqrt[k]{\|\mathbf{C}_\omega^k\|_2}$ asymptotically bounds how much the error is reduced at each step. It is thus often called the *asymptotic convergence rate* and the number of iterations is said to be roughly bounded by $\frac{-\log \varepsilon}{-\log \rho(\mathbf{C}_\omega)}$ (e.g. Hackbusch (2016, Equation 2.31b)). However, while it is tempting to use this as our upper bound $U$, in fact it may not upper bound the number of iterations at all, since $\mathbf{C}_\omega$ is not normal and so in-practice the iteration often goes through a transient phase where the residual norm first *increases* before decreasing (Trefethen & Embree, 2005, Figure 25.6).

Thus we must either take a different approach or make some assumptions. Note that one can in-fact show an $\omega$-dependent, finite-time convergence bound for SOR via the energy norm (Hackbusch,

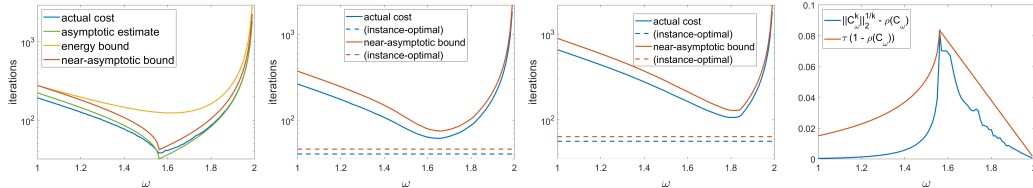

Figure 1: **Left:** comparison of different cost estimates. **Center-left:** mean performance of different parameters across forty instances of form $\mathbf{A} + \frac{12c-3}{20}\mathbf{I}_n$, where $c \sim \text{Beta}(2,6)$. **Center-right:** the same but for $c \sim \text{Beta}(1/2, 3/2)$, which is relatively higher-variance. In both cases the dashed line indicates instance-optimal performance, the matrix $\mathbf{A}$ is a discrete Laplacian of a $100 \times 100$ square domain, and the targets $\mathbf{b}$ are truncated Gaussians. **Right:** asymptocity as measured by the difference between the spectral norm at iteration $k$ and the spectral radius, together with its upper bound $\tau(1 - \rho(\mathbf{C}_\omega))$.

2016, Corollary 3.45), but this can give rather loose upper bounds on the number of iterations (c.f. Figure 1 (left)). Instead, we make the following assumption, which roughly states that convergence always occurs *near* the asymptotic regime, where nearness is measured by a parameter $\tau \in (0, 1)$:

**Assumption 2.1.** *There exists $\tau \in (0, 1)$ s.t. $\forall \ \omega \in \Omega$ the matrix $\mathbf{C}_\omega = \mathbf{I}_n - \mathbf{A}(\mathbf{D}/\omega + \mathbf{L})^{-1}$ satisfies $\|\mathbf{C}_\omega^k\|_2 \leq (\rho(\mathbf{C}_\omega) + \tau(1 - \rho(\mathbf{C}_\omega)))^k$ at $k = \min_{\|\mathbf{C}_\omega^{i+1}\mathbf{b}\|_2 < \varepsilon\|\mathbf{b}\|_2} i$.*

This effectively assumes an upper bound $\rho(\mathbf{C}_\omega) + \tau(1 - \rho(\mathbf{C}_\omega))$ on the empirically observed convergence rate, which gives us a measure of the quality of each parameter $\omega$ for the given instance $(\mathbf{A}, \mathbf{b})$. Note that the specific form of the surrogate convergence rate was chosen both because it is convenient mathematically—it is a convex combination of 1 and the asymptotic rate $\rho(\mathbf{C}_\omega)$—and because empirically we found the degree of "asymptocity" as measured by $\|\mathbf{C}_\omega^k\|_2^{1/k} - \rho(\mathbf{C}_\omega)$ for $k$ right before convergence to vary reasonably similarly to a fraction of $1 - \rho(\mathbf{C}_\omega)$ (c.f. Figure 1 (right)). This makes intuitive sense, as the parameters $\omega$ for which convergence is fastest have the least time to reach the asymptotic regime. Finally, note that since $\lim_{k\to\infty} \|\mathbf{C}_\omega^k\|_2^{1/k} = \rho(\mathbf{C}_\omega)$, for every $\gamma > 0$ there always exists $k'$ s.t. $\|\mathbf{C}_\omega^k\|_2 \leq (\rho(\mathbf{C}_\omega) + \gamma)^k \ \forall \ k \geq k'$; therefore, since $1 - \rho(\mathbf{C}_\omega) > 0$, we view Assumption 2.1 not as a restriction on $\mathbf{C}_\omega$ (and thus on $\mathbf{A}$), but rather as an an assumption on $\varepsilon$ and $\mathbf{b}$. Specifically, the former should be small enough that $\mathbf{C}_\omega^i$ reaches that asymptotic regime for some $i$ before the criterion $\|\mathbf{C}_\omega^k\mathbf{b}\|_2 \leq \varepsilon\|\mathbf{b}\|_2$ is met; for similar reasons, the latter should not happen to be an eigenvector corresponding to a tiny eigenvalue of $\mathbf{C}_\omega$ (c.f. Figure 2 (left)).

Having established this surrogate of the spectral radius, we can use it to obtain a reasonably tight upper bound $U$ on the cost (c.f. Figure 1 (left)). Crucially for learning, we can also establish the following properties via the functional form of $\rho(\mathbf{C}_\omega)$ derived by Young (1971):

**Lemma 2.1.** *Define $U(\omega) = 1 + \frac{-\log \varepsilon}{-\log(\rho(\mathbf{C}_\omega) + \tau(1 - \rho(\mathbf{C}_\omega)))}$, $\alpha = \tau + (1 - \tau)\max\{\beta^2, \omega_{\max} - 1\}$, and $\omega^* = 1 + \beta^2/(1 + \sqrt{1 - \beta^2})^2$, where $\beta = \rho(\mathbf{I}_n - \mathbf{D}^{-1}\mathbf{A})$. Then the following holds:*

*1. $U$ bounds the number of iterations and is itself bounded: $\text{SOR}(\mathbf{A}, \mathbf{b}, \omega) < U(\omega) \leq 1 + \frac{-\log \varepsilon}{-\log \alpha}$*

*2. $U$ is decreasing towards $\omega^*$, and $\frac{-(1-\tau)\log \varepsilon}{\alpha \log^2 \alpha}$-Lipschitz on $\omega \geq \omega^*$ if $\tau \geq \frac{1}{e^2}$ or $\beta^2 \geq \frac{4}{e^2}\left(1 - \frac{1}{e^2}\right)$*

Lemma 2.1 introduces a quantity $\alpha = \tau + (1-\tau)\max\{\beta^2, \omega_{\max} - 1\}$ that appears in the upper bounds on $U(\omega)$ and in its Lipschitz constant. This quantity will in some sense measure the difficulty of learning: if $\alpha$ is close to 1 for many of the instances under consideration then learning will be harder. Crucially, all quantities in the result are spectral and do not depend on the dimensionality of the matrix.

## 2.3 Performing as well as the best fixed $\omega$

Having shown these properties of $U$, we now show that it is learnable via Tsallis-INF (Abernethy et al., 2015; Zimmert & Seldin, 2021), a bandit algorithm which at each instance $t$ samples $\omega_t$ from a discrete probability distribution over a grid of $d$ relaxation parameters, runs SOR with $\omega_t$ on the linear system $(\mathbf{A}_t, \mathbf{b}_t)$, and uses the number of iterations required $\text{SOR}_t(\omega_t)$ as feedback to update the probability distribution over the grid. The scheme is described in full in Algorithm 2. Note that it is a relative of the simpler and more familiar Exp3 algorithm (Auer et al., 2002), but has a slightly better dependence on the grid size $d$. In Theorem 2.1, we bound the cost of using the parameters $\omega_t$ suggested by Tsallis-INF by the total cost of using the best fixed parameter $\omega \in \Omega$ at all iterations—as measured by the surrogate bounds $U_t$—plus a term that increases sublinearly in $T$ and a term that decreases in the size of the grid.

**Algorithm 2:** Online tuning of a linear system solver using Tsallis-INF. The probabilities can be computed using Newton's method (e.g. Zimmert & Seldin (2021, Algorithm 2)).

**Input:** solver $\texttt{SOLVE} : \mathbb{R}^{n \times n} \times \mathbb{R}^n \times \Omega \mapsto \mathbb{Z}_{>0}$, instance sequence $\{(\mathbf{A}_t, \mathbf{b}_t)\}_{t=1}^T \subset \mathbb{R}^{n \times n} \times \mathbb{R}^n$, normalization $K > 0$, parameter grid $\mathbf{g} \in \Omega^d$, step-size $\eta > 0$

$\mathbf{k} \leftarrow \mathbf{0}_d$                         `// initialize vector of cumulative costs`

**for** $t = 1, \ldots, T$ **do**

    $\mathbf{p} \leftarrow \arg\min_{\mathbf{p} \in \triangle_d} \langle \mathbf{k}, \mathbf{p} \rangle - \frac{4K}{\eta} \sum_{i=1}^d \sqrt{\mathbf{p}_{[i]}}$        `// compute probabilities`

    sample $i_t \in [d]$ w.p. $\mathbf{p}_{[i_t]}$ and set $\omega_t = \mathbf{g}_{[i_t]}$      `// sample action from grid`

    $\mathbf{k}_{[i_t]} \leftarrow \mathbf{k}_{[i_t]} + (\texttt{SOLVE}(\mathbf{A}_t, \mathbf{b}_t, \omega_t) - 1)/\mathbf{p}_{[i_t]}$  `// run solver and update cost`

---

**Theorem 2.1.** *Define $\alpha_t = \tau_t + (1 - \tau_t) \max\{\beta_t^2, \omega_{\max} - 1\}$, where $\beta_t = \rho(\mathbf{I}_n - \mathbf{D}_t^{-1}\mathbf{A}_t)$ and $\tau_t$ is the minimal $\tau$ satisfying Assumption 2.1 and the second part of Lemma 2.1. If we run Algorithm 2 using SOR initialized at $\mathbf{x} = \mathbf{0}_n$ as the solver, $\mathbf{g}_{[i]} = 1 + (\omega_{\max} - 1)\frac{i}{d}$ as the parameter grid, normalization $K \geq \frac{-\log \varepsilon}{-\log \alpha_{\max}}$ for $\alpha_{\max} = \max_t \alpha_t$, and step-size $\eta = 1/\sqrt{T}$ then the expected number of iterations is bounded as*

$$\mathbb{E} \sum_{t=1}^T SOR_t(\omega_t) \leq 2K\sqrt{2dT} + \sum_{t=1}^T \frac{-\log \varepsilon}{d \log^2 \alpha_t} + \min_{\omega \in (0, \omega_{\max}]} \sum_{t=1}^T U_t(\omega) \tag{4}$$

*Using $\omega_{\max} = 1 + \max_t \left( \frac{\beta_t}{1 + \sqrt{1 - \beta_t^2}} \right)^2$, $K = \frac{-\log \varepsilon}{-\log \alpha_{\max}}$, and $d = \sqrt[3]{\frac{T}{2} \bar{\gamma}^2 \log^2 \alpha_{\max}}$, for $\bar{\gamma} = \frac{1}{T} \sum_{t=1}^T \frac{1}{\log^2 \alpha_t}$, yields*

$$\mathbb{E} \sum_{t=1}^T SOR_t(\omega_t) \leq 3 \log \frac{1}{\varepsilon} \sqrt[3]{\frac{2\bar{\gamma}T^2}{\log^2 \alpha_{\max}}} + \min_{\omega \in (0,2)} \sum_{t=1}^T U_t(\omega) \leq 3 \log \frac{1}{\varepsilon} \sqrt[3]{\frac{2T^2}{\log^4 \alpha_{\max}}} + \min_{\omega \in (0,2)} \sum_{t=1}^T U_t(\omega) \tag{5}$$

Thus *asymptotically* (as $T \to \infty$) the *average* cost on each instance is that of the best fixed $\omega \in (0, 2)$, as measured by the surrogate loss functions $U_t(\omega)$. The result clearly shows that the difficulty of the learning problem can be measured by how close the values of $\alpha_t$ are to one. As a quantitative example, for the somewhat "easy" case of $\tau_t \leq 0.2$ and $\beta_t \leq 0.9$, the first term is $< T \log \frac{1}{\varepsilon}$—i.e. we take at most $\log \frac{1}{\varepsilon}$ excess iterations on average—after around 73K instances.

The proof of Theorem 2.1 (c.f. Section E) takes advantage of the fact that the upper bounds $U_t$ are always decreasing wherever they are not locally Lipschitz; thus for any $\omega \in (0, \omega_{\max}]$ the next highest grid value in $\mathbf{g}$ will either be better or $\mathcal{O}(1/d)$ worse. This allows us to obtain the same $\mathcal{O}(T^{2/3})$ rate as the optimal Lipschitz-bandit regret (Kleinberg, 2004), despite $U_t$ being only semi-Lipschitz. One important note is that setting $\omega_{\max}$, $K$, and $d$ to obtain this rate involves knowing bounds on spectral properties of the instances. The optimal $\omega_{\max}$ requires a bound on $\max_t \beta_t$ akin to that used by solvers like Chebyshev semi-iteration; assuming this and a reasonable sense of how many iterations are typically required is enough to estimate $\alpha_{\max}$ and then set $d = \sqrt[3]{\frac{T/2}{\log^2 \alpha_{\max}}}$, yielding the right-hand bound in (5). Lastly, we note that Tsallis-INF adds quite little computational overhead: it has a per-instance update cost of $\mathcal{O}(d)$, which for $d = \mathcal{O}(\sqrt[3]{T})$ is likely to be negligible in practice.

### 2.4 THE DIAGONALLY SHIFTED SETTING

The previous analysis is useful when a fixed $\omega$ is good for most instances $(\mathbf{A}_t, \mathbf{b}_t)$. A non-fixed comparator can have much stronger performance (c.f. the dashed lines in Figure 1 (center)), so in this section we study how to use additional, known structure in the form of diagonal shifts: at all $t \in [T]$, $\mathbf{A}_t = \mathbf{A} + c_t \mathbf{I}_n$ for some fixed $\mathbf{A}$ and scalar $c_t$. It is easy to see that selecting instance-dependent $\omega_t$ using the value of the shift is exactly the *contextual bandit* setting (Beygelzimer et al., 2011), in which the comparator is a fixed *policy* $f : \mathbb{R} \mapsto \Omega$ that maps the given scalars to parameters for them. Here the regret is defined by $\text{Regret}_f(\{\ell_t\}_{t=1}^T) = \sum_{t=1}^T \ell_t(\omega_t) - \sum_{t=1}^T \ell_t(f(c_t))$. Notably, if $f$ is the optimal mapping from $c_t$ to $\omega$ then sublinear regret implies doing nearly optimally at every instance. In our case, the policy $\omega^*$ minimizing $U_t$ is a well-defined function of $\mathbf{A}_t$ (c.f. Lemma 2.1) and thus of $c_t$ (Young, 1971); in fact, we can show that the policy is Lipschitz w.r.t. $c_t$ (c.f. Lemma E.1). This allows us to use a very simple algorithm—discretizing the space of offsets $c_t$ into $m$ intervals and running Tsallis-INF separately on each—to obtain $\mathcal{O}(T^{3/4})$ regret w.r.t. the instance-optimal policy $\omega^*$:

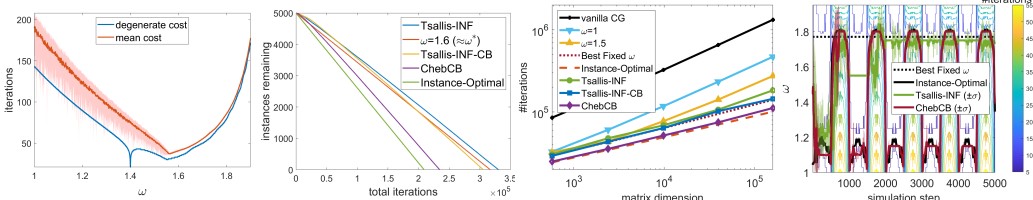

Figure 2: **Left:** solver cost for $\mathbf{b}$ drawn from a truncated Gaussian v.s. $\mathbf{b}$ a small eigenvector of $\mathbf{C}_{1.4}$. **Center-left:** cost to solve 5K diagonally shifted systems $\mathbf{A}_t = \mathbf{A} + \frac{12c_t-3}{20}\mathbf{I}_n$ for $c_t \sim \text{Beta}(2,6)$. **Center-right:** total SSOR-preconditioned CG iterations taken while solving the 2D heat equation with a time-varying diffusion coefficient (used as context) on different grids, as a function of the linear system dimension. **Right:** (smoothed) parameters chosen at each timestep of one such simulation, overlaid on a contour plot of the cost of solving the system at step $t$ with parameter $\omega$ (c.f. Appendix G).

**Theorem 2.2** (c.f. Theorem E.1). *Suppose all offsets $c_t$ lie in $[c_{\min}, c_{\min} + C]$ for some $c_{\min} > -\lambda_{\min}(\mathbf{A})$, and define $L = \frac{1+\beta_{\max}}{\beta_{\max}\sqrt{1-\beta_{\max}^2}}\left(\frac{\lambda_{\min}(\mathbf{D})+c_{\min}+1}{\lambda_{\min}(\mathbf{D})+c_{\min}}\right)^2$ for $\beta_{\max}$ as in Theorem 2.1. Then there is a discretization of this interval s.t. running Algorithm 2 separately on each sequence of contexts in each bin with appropriate parameters results in expected cost*

$$\mathbb{E}\sum_{t=1}^{T} SOR_t(\omega_t) \leq \sqrt[4]{\frac{54C^3L^3T}{\log^2\alpha_{\max}}} + \frac{4\log\frac{1}{\varepsilon}}{\log\frac{1}{\alpha_{\max}}}\sqrt[4]{24CLT^3} + \sum_{t=1}^{T} U_t(\omega^*(c_t)) \tag{6}$$

Observe that, in addition to $\alpha_t$, the difficulty of this learning problem also depends on the maximum spectral radius $\beta_{\max}$ of the Jacobi matrices $\mathbf{I}_n - \mathbf{D}^{-1}(c_t)\mathbf{A}(c_t)$ via the Lipschitz constant $L$ of $\omega^*$.

## 2.5 TUNING PRECONDITIONED CONJUGATE GRADIENT

CG is perhaps the most-used solver for positive definite systems; while it can be run without tuning, in practice significant acceleration can be realized via a good preconditioner such as (symmetric) SOR. The effect of $\omega$ on CG performance can be somewhat distinct from that of regular SOR, requiring a separate analysis. We use the condition number analysis of Axelsson (1994, Theorem 7.17) to obtain an upper bound $U^{\text{CG}}(\omega)$ on the number of iterations required $\text{CG}(\mathbf{A}, \mathbf{b}, \omega)$ to solve a system. While the resulting bounds match the shape of the true performance less exactly than the SOR bounds (c.f. Figure 4), they still provide a somewhat reasonable surrogate. After showing that these functions are also semi-Lipschitz (c.f. Lemma E.2), we can bound the cost of tuning CG using Tsallis-INF:

**Theorem 2.3.** *Set $\mu_t = \rho(\mathbf{D}_t\mathbf{A}_t^{-1})$, $\mu_{\max} = \max_t \mu_t$, $\overline{\sqrt{\mu}} = \frac{1}{T}\sum_{t=1}^{T}\sqrt{\mu}_t$, and $\kappa_{\max} = \max_t \kappa(\mathbf{A}_t)$. If $\min_t \mu_t - 1$ is a positive constant then for Algorithm 2 using preconditioned CG as the solver there exists a parameter grid $\mathbf{g} \in [2\sqrt{2}+2, \omega_{\max}]^d$ and normalization $K > 0$ such that*

$$\mathbb{E}\sum_{t=1}^{T} CG_t(\omega_t) = \mathcal{O}\left(\sqrt[3]{\frac{\log^2\frac{\sqrt{\kappa_{\max}}}{\varepsilon}}{\log^2\frac{\sqrt{\mu_{\max}}-1}{\sqrt{\mu_{\max}}+1}}\overline{\sqrt{\mu}}T^2}\right) + \min_{\omega\in(0,2)}\sum_{t=1}^{T} U_t(\omega) \tag{7}$$

Observe that the rate in $T$ remains the same as for SOR, but the difficulty of learning now scales mainly with the spectral radii of the matrices $\mathbf{D}_t\mathbf{A}_t^{-1}$.

## 3 A STOCHASTIC ANALYSIS OF SYMMETRIC SOR

Assumption 2.1 in the previous section effectively encodes the idea that convergence will not be too quick for a typical target vector $\mathbf{b}$, e.g. it will not be a low-eigenvalue eigenvector of $\mathbf{C}_\omega$ for some otherwise suboptimal $\omega$ (e.g. Figure 2 (left)). Another way of staying in a "typical regime" is randomness, which is what we assume in this section. Specifically, we assume that $\mathbf{b}_t = m_t\mathbf{u}_t \; \forall t \in [T]$, where $\mathbf{u}_t \in \mathbb{R}^n$ is uniform on the unit sphere and $m_t^2$ is a $\chi^2$ random variable with $n$ degrees of freedom truncated to $[0, n]$. Since the standard $n$-dimensional Gaussian is exactly the case of untruncated $m_t^2$, $\mathbf{b}$ can be described as coming from a radially truncated normal distribution. Note also that the exact choice of truncation was done for convenience; any finite bound $\geq n$ yields similar results.

We also make two other changes: (1) we study *symmetric* SOR (SSOR) and (2) we use an absolute convergence criterion, i.e. $\|\mathbf{r}_k\|_2 \leq \varepsilon$, not $\|\mathbf{r}_k\|_2 \leq \varepsilon\|\mathbf{r}_0\|_2$. Symmetric SOR (c.f. Algorithm 8) is very

similar to the original, except the linear system being solved at every step is now symmetric: $\breve{\mathbf{W}}_\omega = \frac{\omega}{2-\omega}\mathbf{W}_\omega \mathbf{D}^{-1}\mathbf{W}_\omega^T$. Note that the defect reduction matrix $\breve{\mathbf{C}}_\omega = \mathbf{I}_n - \mathbf{A}\breve{\mathbf{W}}_\omega^{-1}$ is still not normal, but it *is* (non-orthogonally) similar to a symmetric matrix, $\mathbf{A}^{-1/2}\breve{\mathbf{C}}_\omega \mathbf{A}^{1/2}$. SSOR is twice as expensive per-iteration, but often converges in fewer steps, and is commonly used as a base method because of its spectral properties (e.g. by the Chebyshev semi-iteration, c.f. Hackbusch (2016, Section 8.4.1)).

## 3.1 REGULARITY OF THE EXPECTED COST FUNCTION

We can then show that the expected cost $\mathbb{E}_\mathbf{b}\mathsf{SSOR}(\mathbf{A},\mathbf{b},\omega)$ is Lipschitz w.r.t. $\omega$ (c.f. Corollary F.1). Our main idea is the observation that, whenever the error $\|\breve{\mathbf{C}}_\omega^k \mathbf{b}\|_2$ falls below the tolerance $\varepsilon$, randomness should ensure that it does not fall so close to the threshold that the error $\|\breve{\mathbf{C}}_{\omega'}^k \mathbf{b}\|_2$ of a nearby $\omega'$ is not also below $\varepsilon$. Although clearly related to dispersion (Balcan et al., 2018), here we study the behavior of a continuous function around a threshold, rather than the locations of the costs' discontinuities.

Our approach has two ingredients, the first being Lipschitzness of the error $\|\breve{\mathbf{C}}_\omega^k \mathbf{b}\|_2$ at each iteration $k$ w.r.t. $\omega$, which ensures $\|\breve{\mathbf{C}}_{\omega'}^k \mathbf{b}\|_2 \in (\varepsilon, \varepsilon + \mathcal{O}(|\omega - \omega'|)]$ if $\|\breve{\mathbf{C}}_\omega^k \mathbf{b}\|_2 \leq \varepsilon < \|\breve{\mathbf{C}}_{\omega'}^k \mathbf{b}\|_2$. The second ingredient is anti-concentration, specifically that the probability that $\|\breve{\mathbf{C}}_\omega^k \mathbf{b}\|_2$ lands in $(\varepsilon, \varepsilon + \mathcal{O}(|\omega - \omega'|)]$ is $\mathcal{O}(|\omega - \omega'|)$. While intuitive, both steps are made difficult by powering: for high $k$ the random variable $\|\breve{\mathbf{C}}_\omega^k \mathbf{b}\|_2$ is highly concentrated because $\rho(\breve{\mathbf{C}}_\omega) \ll 1$; in fact its measure over the interval is $\mathcal{O}(|\omega - \omega'|/\rho(\breve{\mathbf{C}}_\omega)^k)$. To cancel this, the Lipschitz constant of $\|\breve{\mathbf{C}}_\omega^k \mathbf{b}\|_2$ must scale with $\rho(\breve{\mathbf{C}}_\omega)^k$, which we can show because switching to SSOR makes $\breve{\mathbf{C}}_\omega^k$ is similar to a normal matrix. The other algorithmic modification we make—using absolute rather than relative tolerance—is so that $\|\breve{\mathbf{C}}_\omega^k \mathbf{b}\|_2^2$ is (roughly) a sum of i.i.d. $\chi^2$ random variables; note that the square of relative tolerance criterion $\|\breve{\mathbf{C}}_\omega^k \mathbf{b}\|_2^2/\|\mathbf{b}\|_2^2$ does not admit such a result. At the same time, absolute tolerance does not imply an a.s. bound on the number of iterations if $\|\mathbf{b}\|_2$ is unbounded, which is why we truncate its distribution.

Lipschitzness follows because $|\mathbb{E}_\mathbf{b}\mathsf{SSOR}(\omega) - \mathbb{E}_\mathbf{b}\mathsf{SSOR}(\omega')|$ can be bounded using Jensen's inequality by the probability that $\omega$ and $\omega'$ have different costs $k \neq l$, which is at most the probability that $\|\breve{\mathbf{C}}_\omega^k \mathbf{b}\|_2$ or $\|\breve{\mathbf{C}}_{\omega'}^l \mathbf{b}\|_2$ land in an interval of length $\mathcal{O}(|\omega - \omega'|)$. Note that the Lipschitz bound includes an $\tilde{\mathcal{O}}(\sqrt{n})$ factor, which results from $\breve{\mathbf{C}}_\omega^k$ having stable rank $\ll n$ due to powering. Regularity of $\mathbb{E}_\mathbf{b}\mathsf{SSOR}$ leads directly to regret guarantee for the same algorithm as before, Tsallis-INF:

**Theorem 3.1.** *Define* $\kappa_{\max} = \max_t \kappa(\mathbf{A}_t)$ *to be the largest condition number and* $\beta_{\min} = \min_t \rho(\mathbf{I}_n - \mathbf{D}_t^{-1}\mathbf{A}_t)$. *Then there exists* $K = \Omega(\log \frac{n}{\varepsilon})$ *s.t. running Algorithm 2 with SSOR has regret*

$$\mathbb{E}\sum_{t=1}^T SSOR_t(\omega_t) - \min_{\omega \in [1, \omega_{\max}]}\sum_{t=1}^T SSOR_t(\omega) \leq 2K\sqrt{2dT} + \frac{32K^4 T}{\beta_{\min}^4 d}\sqrt{\frac{2n\kappa_{\max}}{\pi}} \tag{8}$$

Setting $d = \Theta(K^2\sqrt[3]{nT})$ yields a regret bound of $\mathcal{O}(\log^2 \frac{n}{\varepsilon}\sqrt[3]{T^2\sqrt{n}})$. Note that, while this shows convergence to the true optimal parameter, the constants in the regret term are much worse, not just due to the dependence on $n$ but also in the powers of the number of iterations. Thus this result can be viewed as a proof of the asymptotic ($T \to \infty$) correctness of Tsallis-INF for tuning SSOR.

## 3.2 CHEBYSHEV REGRESSION FOR DIAGONAL SHIFTS

For the shifted setting, we can use the same approach to prove that $\mathbb{E}_\mathbf{b}\mathsf{SSOR}(\mathbf{A} + c\mathbf{I}_n, \mathbf{b}, \omega)$ is Lipschitz w.r.t. the diagonal offset $c$ (c.f. Corollary F.2); for $n = O(1)$ this implies regret $\tilde{\mathcal{O}}(T^{3/4}\sqrt{n})$ for the same discretization-based algorithm as in Section 2.4. While optimal for Lipschitz functions, the method does not readily adapt to nice data, leading to various smoothed comparators (Krishnamurthy et al., 2019; Majzoubi et al., 2020; Zhu & Mineiro, 2022); however, as we wish to compete with the true optimal policy, we stay in the original setting and instead highlight how this section's semi-stochastic analysis allows us to study a very different class of bandit algorithms.

In particular, since we are now working directly with the cost function rather than an upper bound, we are able to utilize a more practical regression-oracle algorithm, SquareCB (Foster & Rakhlin, 2020). It assumes a class of regressors $h : [c_{\min}, c_{\min} + C] \times [d] \mapsto [0, 1]$ with at least one function that perfectly predicts the expected performance $\mathbb{E}_\mathbf{b}\mathsf{SSOR}(\mathbf{A} + c\mathbf{I}_n, \mathbf{b}, \mathbf{g}_{[i]})$ of each action $\mathbf{g}_{[i]}$ given the context $c$; a small amount of model misspecification is allowed. If there exists an online algorithm that can obtain low regret w.r.t. this function class, then SquareCB can obtain low regret w.r.t. any policy.

---

**Algorithm 3:** ChebCB: SquareCB with a follow-the-leader oracle and polynomial regressor class.

**Input:** solver $\text{SOLVE} : \mathbb{R}^{n \times n} \times \mathbb{R}^n \times \Omega \mapsto \mathbb{Z}_{>0}$, instance sequence $\{(\mathbf{A}_t, \mathbf{b}_t)\}_{t=1}^T \subset \mathbb{R}^{n \times n} \times \mathbb{R}^n$,
 context sequence $\{c_t\}_{t=1}^T \subset [c_{\min}, c_{\min} + C]$, learning rate $\eta > 0$, parameter grid $\mathbf{g} \in \Omega^d$,
 Chebyshev polynomial features $\mathbf{f} : [c_{\min}, c_{\min} + C] \mapsto \mathbb{R}^{m+1}$, normalizations $K, L, N > 0$

**for** $t = 1, \dots, T$ **do**

$\quad \theta_i \leftarrow \underset{|\theta_{[0]}| \leq \frac{1}{N}, |\theta_{[j]}| \leq \frac{2CL}{KNj}}{\arg\min} \sum_{\substack{s=1 \\ i_s = i}}^{t-1} \left( \langle \theta, \mathbf{f}(c_s) \rangle - \frac{k_s}{KN} \right)^2 \forall i \in [d]$     `// update models`

$\quad \mathbf{s}_{[i]} \leftarrow \langle \theta_i, \mathbf{f}(c_t) \rangle \ \forall i \in [d]$     `// compute model predictions`

$\quad i^* \leftarrow \arg\min_{i \in [d]} \mathbf{s}_{[i]}$

$\quad \mathbf{p}_{[i]} \leftarrow \frac{1}{d + \eta(\mathbf{s}_{[i]} - \mathbf{s}_{[i^*]})} \ \forall i \neq i^*$     `// compute probability of each action`

$\quad \mathbf{p}_{[i^*]} \leftarrow 1 - \sum_{i \neq i^*} \mathbf{p}_{[i]}$

$\quad$ sample $i_t \in [d]$ w.p. $\mathbf{p}_{[i_t]}$ and set $\omega_t = \mathbf{g}_{[i_t]}$     `// sample action`

$\quad k_t \leftarrow \text{SOLVE}(\mathbf{A}_t, \mathbf{b}_t, \omega_t) - 1$     `// run solver and update cost`

---

To apply it we must specify a suitable class of regressors, bound its approximation error, and specify an algorithm attaining low regret over this class. Since $m$ terms of the Chebyshev series suffice to approximate a Lipschitz function with error $\tilde{\mathcal{O}}(1/m)$, we use Chebyshev polynomials in $c$ with learned coefficients—i.e. models $\langle \theta, \mathbf{f}(c) \rangle = \sum_{j=0}^m \theta_{[j]} P_j(c)$, where $P_j$ is the $j$th Chebyshev polynomial—as our regressors for each action. To keep predictions bounded, we add constraints $|\theta_{[j]}| = \mathcal{O}(1/j)$, which we can do without losing approximation power due to the decay of Chebyshev series coefficients. This allows us to show $\mathcal{O}(dm \log T)$ regret for Follow-The-Leader via Hazan et al. (2007, Theorem 5) and then apply Foster & Rakhlin (2020, Theorem 5) to obtain the following guarantee:

**Theorem 3.2** (Corollary of Theorem C.4). *Suppose $c_{\min} > -\lambda_{\min}(\mathbf{A})$. Then Algorithm 3 with appropriate parameters has regret w.r.t. any policy $f : [c_{\min}, c_{\min} + C] \mapsto \Omega$ of*

$$\mathbb{E} \sum_{t=1}^T SSOR_t(\omega_t) - \sum_{t=1}^T SSOR_t(f(c_t)) \leq \tilde{\mathcal{O}} \left( d\sqrt{mnT} + \frac{T\sqrt{dn}}{m} + \frac{T\sqrt{n}}{d} \right) \tag{9}$$

Setting $d = \Theta(T^{2/11})$ and $m = \Theta(T^{3/11})$ yields $\tilde{\mathcal{O}}(T^{9/11}\sqrt{n})$ regret, so we asymptotically attain instance-optimal performance, albeit at a rather slow rate. The rate in $n$ is also worse than e.g. our semi-stochastic result for comparing to a fixed $\omega$ (c.f. Theorem 3.1), although to obtain this the latter algorithm uses $d = \mathcal{O}(\sqrt[3]{n})$ grid points, making its overhead nontrivial. We compare ChebCB to the Section 2.4 algorithm based on Tsallis-INF (among other methods), and find that, despite the former's worse guarantees, it seems able to converge to an instance-optimal policy much faster than the latter.

## 4 CONCLUSION AND LIMITATIONS

We have shown that bandit algorithms provably learn to parameterize SOR, an iterative linear system solver, and do as well asymptotically as the best fixed $\omega$ in terms of either (a) a near-asymptotic measure of cost or (b) expected cost. We further show that a modern *contextual* bandit method attains near-instance-optimal performance. Both procedures require only the iteration count as feedback and have limited computational overhead settings, making them practical to deploy. Furthermore, the theoretical ideas in this work—especially the use of contextual bandits for taking advantage of instance structure and Section 3.1's conversion of anti-concentrated Lipschitz criteria to Lipschitz expected costs—have the strong potential to be applicable to other domains of data-driven algorithm design.

At the same time, only the near-asymptotic results yield reasonable bound on the instances needed to attain good performance, with the rest having large spectral and dimension-dependent factors; the latter is the most obvious area for improvement. Furthermore, the near-asymptotic upper bounds are somewhat loose for sub-optimal $\omega$ and for preconditioned CG, and as discussed in Section 2.4 do not seem amenable to regression-based CB. Beyond this, a natural direction is to attain semi-stochastic results for non-stationary solvers like preconditioned CG, or either type of result for the many other algorithms in scientific computing. Practically speaking, work on multiple parameters—e.g. the spectral bounds used for Chebyshev semi-iteration, or multiple relaxation parameters for Block-SOR—would likely be most useful. A final direction is to design online learning algorithms that exploit properties of the losses beyond Lipschitzness, or CB algorithms that take better advantage of such functions.

ACKNOWLEDGMENTS

We thank Akshay Krishnamurthy and Ainesh Bakshi for helpful feedback. This work was supported in part by National Science Foundation grants IIS-1705121, IIS-1838017, IIS-1901403, IIS-2046613, IIS-2112471, and OAC-2203821, the Defense Advanced Research Projects Agency under cooperative agreement HR00112020003, a TCS Presidential Fellowship, and funding from Meta, Morgan Stanley, Amazon, Google, and Jane Street. Any opinions, findings and conclusions or recommendations expressed in this material are those of the author(s) and do not necessarily reflect the views of any of these funding agencies.

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

# A  RELATED WORK AND COMPARISONS

Our analysis falls mainly into the framework of data-driven algorithm design, which has a long history (Gupta & Roughgarden, 2017; Balcan, 2021). Closely related is the study by Gupta & Roughgarden (2017) of the sample complexity of learning the step-size of gradient descent, which can also be used to solve linear systems. While their sample complexity guarantee is logarithmic in the precision $1/\varepsilon$, directly applying their Lipschitz-like analysis in a bandit setting yields regret with a polynomial dependence; note that a typical setting of $\varepsilon$ is $10^{-8}$. Mathematically, their analysis relies crucially on the iteration reducing error at every step, which is well-known *not* to be the case for SOR (e.g. Trefethen & Embree (2005, Figure 25.6)). Data-driven numerical linear algebra was studied most explicitly by Bartlett et al. (2022), who provided sample complexity framework applicable to many algorithms; their focus is on the offline setting where an algorithm is learned from a batch of samples. While they do not consider linear systems directly, in Appendix A.1 we do compare to the guarantee their framework implies for SOR; we obtain similar sample complexity with an efficient learning procedure, at the cost of a strong distributional assumption on the target vector. Note that generalization guarantees have been shown for convex quadratic programming—which subsumes linear systems— by Sambharya et al. (2023); they focus on learning-to-initialize, which we do not consider because for high precisions the initialization quality usually does not have a strong impact on cost. Note that all of the above work also does not provide end-to-end guarantees, only e.g. sample complexity bounds.

Online learning guarantees were shown for the related problem of tuning regularized regression by Balcan et al. (2022), albeit in the easier full information setting and with the target of reducing error rather than computation. Their approach relies on the dispersion technique (Balcan et al., 2018), which often involves showing that discontinuities in the cost are defined by bounded-degree polynomials (Balcan et al., 2020). While possibly applicable in our setting, we suspect using it would lead to unacceptably high dependence on the dimension and precision, as the power of the polynomials defining our decision boundaries is $\mathcal{O}(n^{-\log \varepsilon})$. Lastly, we believe our work is notable within this field as a first example of using contextual bandits, and in doing so competing with the provably instance-optimal policy.

Iterative (discrete) optimization has been studied in the related area of learning-augmented algorithms (a.k.a. algorithms with predictions) (Dinitz et al., 2021; Chen et al., 2022; Sakaue & Oki, 2022), which shows data-dependent performance guarantees as a function of (learned) predictions (Mitzenmacher & Vassilvitskii, 2021); these can then be used as surrogate losses for learning (Khodak et al., 2022). Our construction of an upper bound under asymptotic convergence is inspired by this, although unlike previous work we do not assume access to the bound directly because it depends on hard-to-compute spectral properties. Algorithms with predictions often involve initializing a computation with a prediction of its outcome, e.g. a vector near the solution $\mathbf{A}^{-1}\mathbf{b}$; we do not consider this because the runtime of SOR and other solvers depends fairly weakly on the distance to the initialization.

A last theoretical area is that of gradient-based meta-learning, which studies how to initialize and tune other parameters of gradient descent and related methods (Khodak et al., 2019; Denevi et al., 2019; Saunshi et al., 2020; Chen & Hazan, 2023). This field focuses on learning-theoretic notions of cost such as regret or statistical risk. Furthermore, their guarantees are usually on the error after a fixed number of gradient steps rather than the number of iterations required to converge; targeting the former can be highly suboptimal in scientific computing applications (Arisaka & Li, 2023). This latter work, which connects meta-learning and data-driven scientific computing, analyzes specific case studies for accelerating numerical solvers, whereas we focus on a general learning guarantee.

Empirically, there are many learned solvers (Luz et al., 2020; Taghibakhshi et al., 2021; Li et al., 2023) and even full simulation replacements (Karniadakis et al., 2021; Li et al., 2021); to our knowledge, theoretical studies of the latter have focused on expressivity (Marwah et al., 2021). Amortizing the cost on future simulations (Amos, 2023), these approaches use offline computation to train models that integrate directly with solvers or avoid solving linear systems altogether. In contrast, the methods we propose are online and lightweight, both computationally and in terms of implementation; unlike many deep learning approaches, the additional computation scales slowly with dimension and needs only black-box access to existing solvers. As a result, our methods can be viewed as reasonable baselines, and we discuss an indirect comparison with the CG-preconditioner-learning approach of Li et al. (2023) in Appendix G. Finally, note that improving the performance of linear solvers across a sequence of related instances has seen a lot of study in the scientific computing literature (Parks et al., 2006; Tebbens & Tůma, 2007; Elbouyahyaoui et al., 2021). To our knowledge, this work does not give explicit guarantees on the number of iterations, and so a direct theoretical comparison is challenging.

While not the focus of our work, we briefly note the generalization implications of our semi-stochastic analysis. Suppose for any $\alpha > 0$ we have $T = \tilde{\mathcal{O}}(\frac{1}{\alpha^2}\text{polylog}\frac{n}{\delta})$ i.i.d. samples from a distribution $\mathcal{D}$ over matrices $\mathbf{A}_t$ satisfying the assumptions in Section 2.1 and truncated Gaussian targets $\mathbf{b}_t$. Then empirical risk minimization $\hat{\omega} = \arg\min_{\hat{\omega} \in \mathbf{g}} \sum_{t=1}^{T} \text{SSOR}(\mathbf{A}_t, \mathbf{b}_t, \omega)$ over a uniform grid $\mathbf{g} \in [1, \omega_{\max}]^d$ of size $d = \tilde{\mathcal{O}}(\sqrt{nT})$ will be $\alpha$-suboptimal w.p. $\geq 1 - \delta$:

**Corollary A.1.** *Let $\mathcal{D}$ be a distribution over matrix-vector pairs $(\mathbf{A}, \mathbf{b}) \in \mathbb{R}^{n \times n} \times \mathbb{R}^n$ where $\mathbf{A}$ satisfies the SOR conditions and for every $\mathbf{A}$ the conditional distribution of $\mathcal{D}$ given $\mathbf{A}$ over $\mathbb{R}^n$ is the truncated Gaussian. For every $T \geq 1$ consider the algorithm that draws $T$ samples $(\mathbf{A}_t, \mathbf{b}_t) \sim \mathcal{D}$ and outputs $\hat{\omega} = \arg\min_{\hat{\omega} \in \mathbf{g}} \sum_{t=1}^{T} SSOR_t(\omega)$, where $\mathbf{g}_{[i]} = 1 + (\omega_{\max} - 1)\frac{i-1/2}{d}$ and $d = \frac{L\sqrt{T}}{K}$ for $L$ as in Corollary F.1. Then $T = \tilde{\mathcal{O}}\left(\frac{1}{\alpha^2}\text{polylog}\frac{n}{\varepsilon\delta}\right)$ samples suffice to ensure $\mathbb{E}_{\mathcal{D}} SSOR(\mathbf{A}, \mathbf{b}, \hat{\omega}) \leq \min_{\omega \in [1,\omega_{\max}]} SSOR(\mathbf{A}, \mathbf{b}, \omega) + \alpha$ holds w.p. $\geq 1 - \delta$.*

*Proof.* A standard covering bound (see e.g. Lafferty et al. (2010, Theorem 7.82)) followed by an application of Corollary F.1 implies that w.p. $\geq 1 - \delta$

$$
\begin{aligned}
\mathbb{E}_{\mathcal{D}} SSOR(\mathbf{A}, \mathbf{b}, \hat{\omega}) &\leq \min_{\omega \in \mathbf{g}} \mathbb{E}_{\mathcal{D}} SSOR(\mathbf{A}, \mathbf{b}, \omega) + 3K\sqrt{\frac{2}{T}\log\frac{2d}{\delta}} \\
&= \min_{\omega \in \mathbf{g}} \mathbb{E}_{\mathbf{A}}[\mathbb{E}_{\mathbf{b}} SSOR(\mathbf{A}, \mathbf{b}, \omega) | \mathbf{A}] + 3K\sqrt{\frac{2}{T}\log\frac{2d}{\delta}} \\
&\leq \min_{\omega \in [1,\omega_{\max}]} \mathbb{E}_{\mathbf{A}}\left[\mathbb{E}_{\mathbf{b}} SSOR(\mathbf{A}, \mathbf{b}, \omega) + \frac{L}{d}\bigg|\mathbf{A}\right] + 3K\sqrt{\frac{2}{T}\log\frac{2d}{\delta}} \quad (10) \\
&= \min_{\omega \in [1,\omega_{\max}]} \mathbb{E}_{\mathcal{D}} SSOR(\mathbf{A}, \mathbf{b}, \omega) + \frac{L}{d} + 3K\sqrt{\frac{2}{T}\log\frac{2d}{\delta}} \\
&\leq \min_{\omega \in [1,\omega_{\max}]} \mathbb{E}_{\mathcal{D}} SSOR(\mathbf{A}, \mathbf{b}, \omega) + 4K\sqrt{\frac{2}{T}\log\frac{2LT}{K\delta}}
\end{aligned}
$$

Noting that by Corollary F.1 we have $L = \mathcal{O}(K^4\sqrt{n}) = \mathcal{O}(\sqrt{n}\log^4\frac{n}{\varepsilon})$ yields the result. $\quad\square$

This matches directly applying the GJ framework of Bartlett et al. (2022, Theorem 3.3) to our problem:

**Corollary A.2.** *In the same setting as Corollary A.1 but generalizing the distribution to any one whose target vector support is $\sqrt{n}$-bounded, empirical risk minimization (running $\hat{\omega} = \arg\min_{\omega \in [1,\omega_{\max}]} \sum_{t=1}^{T} SSOR_t(\omega)$) has sample complexity $\tilde{\mathcal{O}}\left(\frac{1}{\alpha^2}\text{polylog}\frac{n}{\varepsilon\delta}\right)$.*

*Proof.* For every $(\mathbf{A}, \mathbf{b})$ pair in the support of $\mathcal{D}$ and any $r \in \mathbb{R}$ it is straightforward to define a GJ algorithm (Bartlett et al., 2022, Definition 3.1) that checks if $SSOR(\mathbf{A}, \mathbf{b}, \omega) > r$ by computing $\|\mathbf{r}_k(\omega)\|_2^2 = \|\check{\mathbf{C}}_\omega^k \mathbf{b}\|_2^2$—a degree $2k$ polynomial—for every $k \leq \lfloor r \rfloor$ and returning "True" if one of them satisfies $\|\mathbf{r}_k(\omega)\|_2^2 \leq \varepsilon^2$ and "False" otherwise (and automatically return "True" for $r \geq K$ and "False" for $r < 1$). Since the degree of this algorithm is at most $2K$, the predicate complexity is at most $K$, and the parameter size is 1, by Bartlett et al. (2022, Theorem 3.3) the pseudodimension of $\{SSOR(\cdot, \cdot, \omega) : \omega \in [1, \omega_{\max}]\}$ is $\mathcal{O}(\log K)$. Using the bounded assumption on the target vector—$SSOR \leq K = \mathcal{O}(\log\frac{n}{\varepsilon})$—completes the proof. $\quad\square$

At the same, recent generalization guarantees for tuning regularization parameters of linear regression by Balcan et al. (2022, Theorem 3.2)—who applied dual function analysis (Balcan et al., 2021)—have a quadratic dependence on the instance dimension. Unlike both results—which use uniform convergence—our bound also uses a (theoretically) efficient learning procedure, at the cost of a strong (but in our view reasonable) distributional assumption on the target vectors.

## A.2 APPROXIMATING THE SPECTRAL RADIUS OF THE JACOBI ITERATION MATRIX

Because the asymptotically optimal $\omega$ is a function of the spectral radius $\beta = \rho(\mathbf{M}_1)$ of the Jacobi iteration matrix, a reasonable baseline is to simply approximate $\beta$ using an eigenvalue solver and then run SOR with the corresponding approximately best $\omega$. It is difficult to compare our results to this approach directly, since the baseline will always run extra matrix iterations while bandit algorithms will asymptotically run no more than the comparator. Furthermore, $\mathbf{M}_1$ is not a normal matrix, a class for which it turns out to be surprisingly difficult to find bounds on the number of iterations required to approximate its largest eigenvalue within some tolerance $\alpha > 0$.

A comparison can be made in the diagonal offset setting by modifying this baseline somewhat and making the assumption that $\mathbf{A}$ has a constant diagonal, so that $\mathbf{M}_1$ is symmetric and we can use randomized block-Krylov to obtain a $\hat{\beta}$ satisfying $|\hat{\beta}^2 - \beta^2| = \mathcal{O}(\varepsilon)$ in $\tilde{\mathcal{O}}(1/\sqrt{\varepsilon})$ iterations w.h.p. (Musco & Musco, 2015, Theorem 1). To modify the baseline, we consider a *preprocessing* algorithm which discretizes $[c_{\min}, c_{\min} + C]$ into $d$ grid points, runs $k$ iterations of randomized block-Krylov on the Jacobi iteration matrix of each matrix $\mathbf{A} + c\mathbf{I}_n$ corresponding to offsets $c$ in this grid, and then for each new offset $c_t$ we set $\omega_t$ using the optimal parameter implied by the approximate spectral radius of the Jacobi iteration matrix of $\mathbf{A} + c\mathbf{I}_n$ corresponding to the closest $c$ in the grid. This algorithm thus does $\tilde{\mathcal{O}}(dk)$ matrix-vector products of preprocessing, and since the upper bounds $U_t$ are $\frac{1}{2}$-Hölder w.r.t. $\omega$ while the optimal policy is Lipschitz w.r.t. $\beta^2$ over an appropriate domain $[1, \omega_{\max}]$ it will w.h.p. use at most $\tilde{\mathcal{O}}(\sqrt{1/k^2 + 1/d})$ more iterations at each step $t \in [T]$ compared to the optimal policy. Thus w.h.p. the total regret compared to the optimal policy $\omega^*$ is

$$\sum_{t=1}^{T} \text{SOR}_t(\omega_t) = \tilde{\mathcal{O}}\left(dk + T/d + T/\sqrt{k}\right) + \sum_{t=1}^{T} U_t(\omega^*(c_t)) \tag{11}$$

Setting $d = \sqrt[4]{T}$ and $k = \sqrt{T}$ yields the rate $\tilde{\mathcal{O}}(T^{3/4})$, which can be compared directly to our $\tilde{\mathcal{O}}(T^{3/4})$ rate for the discretized Tsallis-INF algorithm in Theorem 2.2. The rate of approximating $\rho(\mathbf{M}_1)$ thus matches that of our simplest approach, although unlike the latter (and also unlike ChebCB) it does not guarantee performance as good as the optimal policy in the semi-stochastic setting, where $\omega^*$ might not be optimal. Intuitively, the randomized block-Krylov baseline will also suffer from spending computation on points $c \in [c_{\min}, c_{\min} + C]$ that it does not end up seeing.

**Algorithm 4:** General form of Tsallis-INF. The probabilities can be computed using Newton's method (e.g. Zimmert & Seldin (2021, Algorithm 2)).

---

**Input:** loss sequence $\{\ell_t : [a,b] \mapsto [0,K]\}_{t=1}^T$, action set $\mathbf{g} \in [a,b]^d$, step-sizes $\eta_1, \ldots, \eta_T > 0$

$\mathbf{k} \leftarrow \mathbf{0}_d$         `// initialize vector of cumulative losses`

**for** $t = 1, \ldots, T$ **do**

$\quad\quad \mathbf{p} \leftarrow \arg\min_{\mathbf{p} \in \triangle_d} \langle \mathbf{k}, \mathbf{p} \rangle - \frac{4K}{\eta_t} \sum_{i=1}^d \sqrt{\mathbf{p}_{[i]}}$     `// compute probabilities`

$\quad\quad$ sample $i_t \in [d]$ with probability $\mathbf{p}_{[i_t]}$     `// sample index of an action`

$\quad\quad \mathbf{k}_{[i_t]} \leftarrow \mathbf{k}_{[i_t]} + \ell_t(\mathbf{g}_{[i_t]})/\mathbf{p}_{[i_t]}$     `// play action and update losses`

---

## B    SEMI-LIPSCHITZ BANDITS

We consider a sequence of adaptively chosen loss functions $\ell_1, \ldots, \ell_T : [a,b] \mapsto [0,K]$ on an interval $[a,b] \subset \mathbb{R}$ and upper bounds $u_1, \ldots, u_T : [a,b] \mapsto \mathbb{R}$ satisfying $u_t(x) \geq \ell_t(x) \, \forall \, t \in [T], x \in [a,b]$, where $[T]$ denotes the set of integers from 1 to $T$. Our analysis will focus on the Tsallis-INF algorithm of Abernethy et al. (2015), which we write in its general form in Algorithm 4, although the analysis extends easily to the better-known (but sub-optimal) Exp3 (Auer et al., 2002). For Tsallis-INF, the following two facts follow directly from known results:

**Theorem B.1** (Corollary of Abernethy et al. (2015, Corollary 3.2)). *If $\eta_t = 1/\sqrt{T} \, \forall \, t \in [T]$ then Algorithm 4 has regret $\mathbb{E} \sum_{t=1}^T \ell_t(\mathbf{g}_{[i_t]}) - \min_{i \in [d]} \sum_{t=1}^T \ell_t(\mathbf{g}_{[i]}) \leq 2K\sqrt{2dT}$.*

**Theorem B.2** (Corollary of Zimmert & Seldin (2021, Theorem 1)). *If $\eta_t = 2/\sqrt{t} \, \forall \, t \in [T]$ then Algorithm 4 has regret $\mathbb{E} \sum_{t=1}^T \ell_t(\mathbf{g}_{[i_t]}) - \min_{i \in [d]} \sum_{t=1}^T \ell_t(\mathbf{g}_{[i]}) \leq 4K\sqrt{dT} + 1$.*

We now define a generalization of the Lipschitzness condition that trivially generalizes regular $L$-Lipschitz functions, as well as the notion of *one-sided Lipschitz* functions studied in the stochastic setting by Dütting et al. (2023).

**Definition B.1.** *Given a constant $L \geq 0$ and a point $z \in [a,b]$, we say a function $f : [a,b] \mapsto \mathbb{R}$ is $(L, z)$-**semi-Lipschitz** if $f(x) - f(y) \leq L|x - y| \, \forall \, x, y$ s.t. $|x - z| \leq |y - z|$.*

We now show that Tsallis-INF with bandit access to $\ell_t$ on a discretization of $[a,b]$ attains $\mathcal{O}(T^{2/3})$ regret w.r.t. any fixed $x \in [a,b]$ evaluated by any comparator sequence of semi-Lipschitz upper bounds $u_t$. Note that guarantees for the standard comparator can be recovered by just setting $\ell_t = u_t \, \forall \, t \in [T]$, and that the rate is optimal by Kleinberg (2004, Theorem 4.2).

**Theorem B.3.** *If $u_t \geq \ell_t$ is $(L_t, z)$-semi-Lipschitz $\forall \, t \in [T]$ then Algorithm 4 using action space $\mathbf{g} \in [a,b]^d$ s.t. $\mathbf{g}_{[i]} = a + \frac{b-a}{d}i \, \forall \, i \in [d-1]$ and $\mathbf{g}_{[d]} = z$ has regret*

$$\mathbb{E} \sum_{t=1}^T \ell_t(\mathbf{g}_{[i_t]}) - \min_{x \in [a,b]} \sum_{t=1}^T u_t(x) \leq 2K\sqrt{2dT} + \frac{b-a}{d} \sum_{t=1}^T L_t \tag{12}$$

*Setting $d = \sqrt[3]{\frac{(b-a)^2 \bar{L}^2 T}{2K^2}}$ for $\bar{L} = \frac{1}{T} \sum_{t=1}^T L_t$ yields the bound $3\sqrt[3]{2(b-a)\bar{L}K^2T^2}$.*

*Proof.* Let $\lceil \cdot \rfloor_{\mathbf{g}}$ denote rounding to the closest element of $\mathbf{g}$ in the direction of $z$. Then for $x \in [a,b]$ we have $|\lceil x \rfloor_{\mathbf{g}} - z| \leq |x - z|$ and $|\lceil x \rfloor_{\mathbf{g}} - x| \leq \frac{b-a}{d}$, so applying Theorem B.1 and this fact yields

$$\mathbb{E} \sum_{t=1}^T \ell_t(\mathbf{g}_{[i_t]}) \leq 2K\sqrt{2dT} + \min_{i \in [d]} \sum_{t=1}^T \ell_t(\mathbf{g}_{[i]}) \leq 2K\sqrt{2dT} + \min_{i \in [d]} \sum_{t=1}^T u_t(\mathbf{g}_{[i]})$$

$$= 2K\sqrt{2dT} + \min_{x \in [a,b]} \sum_{t=1}^T u_t(\lceil x \rfloor_{\mathbf{g}})$$

$$\leq 2K\sqrt{2dT} + \frac{b-a}{d} \sum_{t=1}^T L_t + \min_{x \in [a,b]} \sum_{t=1}^T u_t(x) \tag{13}$$

$\square$

**Algorithm 5:** Contextual bandit algorithm using instances of Tsallis-INF over a grid of contexts.

**Input:** loss sequence $\{\ell_t : [a, b] \mapsto [0, K]\}_{t=1}^T$, context sequence $\{c_t\}_{t=1}^T \subset [c, c + C]$,
action set $\mathbf{g} \in [a, b]^d$, discretization $\mathbf{h} \in [c, c + C]^m$
**for** $j = 1, \dots, m$ **do**
    $\mathcal{A}_j = \texttt{Tsallis-INF}(\mathbf{g}, \{\frac{2}{\sqrt{t}}\}_{t=1}^T)$   // start $m$ instances of Algorithm 4
**for** $t = 1, \dots, T$ **do**
    $j_t = \min \arg \min_{j \in [m]} |\mathbf{h}_{[j]} - c_t|$      // pick element of $\mathbf{h}$ closest to $c_t$
    $i_t \leftarrow \mathcal{A}_{j_t}$        // get action from $j_t$th instance of Algorithm 4
    $\ell_t(\mathbf{g}_{[i_t]}) \to \mathcal{A}_{j_t}$       // pass loss to $j_t$th instance of Algorithm 4

For contextual bandits, we restrict to $(L_t, b)$-semi-Lipschitz functions and $L_f$-Lipschitz policies, obtaining $\mathcal{O}(T^{3/4})$ regret; this rate matches known upper and lower bounds for the case where losses are Lipschitz in both actions and contexts (Lu et al., 2010, Theorem 1), although this does not imply optimality of our result.

**Theorem B.4.** *If $u_t \geq \ell_t$ is $(L_t, b)$-semi-Lipschitz and $c_t \in [c, c + C] \; \forall \, t \in [T]$ then Algorithm 5 using action space $\mathbf{g}_{[i]} = a + \frac{b-a}{d} i$ and $\mathbf{h}_{[j]} = c + \frac{C}{m}(j - \frac{1}{2})$ as the grid of contexts has regret w.r.t. any $L_f$-Lipschitz policy $f : [c, c + C] \mapsto [a, b]$ of*

$$\mathbb{E} \sum_{t=1}^T \ell_t(\mathbf{g}_{[i_t]}) - \sum_{t=1}^T u_t(\pi(c_t)) \leq m + 4K\sqrt{dmT} + \left(\frac{CL_f}{m} + \frac{b-a}{d}\right) \sum_{t=1}^T L_t \qquad (14)$$

*Setting $d = \sqrt[4]{\frac{(b-a)^3 \bar{L}^2 T}{4CL_f K^2}}$, $m = \sqrt[4]{\frac{C^3 L_f^3 \bar{L}^2 T}{4(b-a)K^2}}$ yields regret $4\sqrt[4]{4K^2\bar{L}^2(b-a)CL_f T^3} + \sqrt[4]{\frac{C^3 L_f^3 \bar{L}^2 T}{4(b-a)K^2}}$.*

*Proof.* Define $\lceil \cdot \rfloor_{\mathbf{h}}$ to be the operation of rounding to the closest element of $\mathbf{h}$, breaking ties arbitrarily, and set $[T]_j = \{t \in [T] : \lceil c_t \rfloor_{\mathbf{h}} = \mathbf{h}_{[j]}\}$. Furthermore, define $\lceil x \rceil_{\mathbf{g}}$ to be the smallest element $\mathbf{g}_{[i]}$ in $\mathbf{g}$ s.t. $x + \frac{CL_f}{2m} \leq \mathbf{g}_{[i]}$ (or $\max_{i \in [d]} \mathbf{g}_{[i]}$ if such an element does not exist).

$$
\begin{aligned}
\mathbb{E} \sum_{t=1}^T \ell_t(\mathbf{g}_{[i_t]}) &= \mathbb{E} \sum_{j=1}^m \sum_{t \in [T]_j} \ell_t(\mathbf{g}_{[i_t]}) - \min_{i \in [d]} \sum_{t \in [T]_j} \ell_t(\mathbf{g}_{[i]}) + \min_{i \in [d]} \sum_{t \in [T]_j} \ell_t(\mathbf{g}_{[i]}) \\
&\leq m + 4 \sum_{j=1}^m K\sqrt{d|[T]_j|} + \min_{i \in [d]} \sum_{t \in [T]_j} \ell_t(\mathbf{g}_{[i]}) \\
&\leq m + 4K\sqrt{dmT} + \sum_{j=1}^m \min_{i \in [d]} \sum_{t \in [T]_j} u_t(\mathbf{g}_{[i]}) \\
&\leq m + 4K\sqrt{dmT} + \sum_{t=1}^T u_t(\lceil f(\lceil c_t \rfloor_{\mathbf{h}}) \rceil_{\mathbf{g}})
\end{aligned}
\qquad (15)
$$

where the first inequality follows by Theorem B.2, the second applies Jensen's inequality to the left term and $u_t \geq \ell_t$ on the right, and the last uses optimality of each $i$ for each $j$. Now since $f$ is $L_f$-Lipschitz we have by definition of $\lceil \cdot \rfloor_{\mathbf{h}}$ that $|f(c_t) - f(\lceil c_t \rfloor_{\mathbf{h}})| \leq \frac{CL_f}{2m}$. This in turn implies that $f(c_t) \leq \lceil f(\lceil c_t \rfloor_{\mathbf{h}}) \rceil_{\mathbf{g}} \leq f(c_t) + \frac{CL_f}{m} + \frac{b-a}{d}$ by definition of $\mathbf{g}$ and $\lceil \cdot \rceil_{\mathbf{g}}$. Since $u_t$ is $(L_t, b)$-semi-Lipschitz, the result follows. $\qquad \square$

# C CHEBYSHEV REGRESSION FOR CONTEXTUAL BANDITS

## C.1 PRELIMINARIES

We first state a Lipschitz approximation result that is standard but difficult-to-find formally. For all $j \in \mathbb{Z}_{\geq 0}$ we will use $P_j(x) = \cos(j \arccos(x))$ to denote the $j$th Chebyshev polynomial of the first kind.

**Theorem C.1.** *Let $f : [\pm 1] \mapsto [\pm K]$ be a $K$-bounded, $L$-Lipschitz function. Then for each integer $m \geq 0$ there exists $\theta \in \mathbb{R}^{m+1}$ satisfying the following properties:*

*1. $|\theta_{[0]}| \leq K$ and $|\theta_{[j]}| \leq 2L/j \ \forall \ j \in [m]$*

*2. $\max_{x \in [\pm 1]} \left| f(x) - \sum_{j=0}^{m} \theta_{[j]} P_j(x) \right| \leq \frac{\pi + \frac{2}{\pi} \log(2m+1)}{m+1} L$*

*Proof.* Define $\theta_{[0]} = \frac{1}{\pi} \int_{-1}^{1} \frac{f(x)}{\sqrt{1-x^2}} dx$ and for each $j \in [m]$ let $\theta_{[j]} = \frac{2}{\pi} \int_{-1}^{1} \frac{f(x) P_j(x)}{\sqrt{1-x^2}} dx$ be the $j$th Chebyshev coefficient. Since $\int_{-1}^{1} \frac{dx}{\sqrt{1-x^2}} = \pi$ we trivially have $|\theta_{[0]}| \leq K$ and by Trefethen (2008, Theorem 4.2) we also have

$$|\theta_{[j]}| \leq \frac{2}{\pi j} \int_{-1}^{1} \frac{|f'(x)|}{\sqrt{1-x^2}} dx \leq \frac{2L}{\pi j} \int_{-1}^{1} \frac{dx}{\sqrt{1-x^2}} = 2L/j \tag{16}$$

for all $j \in [m]$. This shows the first property. For the second, by Trefethen (2008, Theorem 4.4) we have that

$$\max_{x \in [-1,1]} \left| f(x) - \sum_{j=0}^{m} \theta_{[j]} P_j(x) \right| \leq \left( 2 + \frac{4 \log(2m+1)}{\pi^2} \right) \max_{x \in [\pm 1]} |f(x) - p_m^*(x)|$$

$$\leq \left( 2 + \frac{4 \log(2m+1)}{\pi^2} \right) \frac{L\pi}{2(m+1)} = \frac{\pi + \frac{2}{\pi} \log(2m+1)}{m+1} L \tag{17}$$

where $p_m^*$ is the (at most) $m$-degree algebraic polynomial that best approximates $f$ on $[\pm 1]$ and the second inequality is Jackson's theorem (Cheney, 1982, page 147). $\square$

**Corollary C.1.** *Let $f : [a, b] \mapsto [\pm K]$ be a $K$-bounded, $L$-Lipschitz function on the interval $[a, b]$. Then for each integer $m \geq 0$ there exists $\theta \in \mathbb{R}^{m+1}$ satisfying the following properties:*

*1. $|\theta_{[0]}| \leq K$ and $|\theta_{[j]}| \leq \frac{L(b-a)}{j}$*

*2. $\max_{x \in [a,b]} \left| f(x) - \sum_{j=0}^{m} \theta_{[j]} P_j(\frac{2}{b-a}(x-a) - 1) \right| \leq \frac{\pi + \frac{2}{\pi} \log(2m+1)}{2(m+1)} L(b-a)$*

*Proof.* Define $g(x) = f(\frac{b-a}{2}(x+1) + a)$, so that $g : [\pm 1] \mapsto [\pm K]$ is $K$-bounded and $L\frac{b-a}{2}$-Lipschitz. Applying Theorem C.1 yields the result. $\square$

We next state regret guarantees for the SquareCB algorithm of Foster & Rakhlin (2020) in the non-realizable setting:

**Theorem C.2** (Foster & Rakhlin (2020, Theorem 5)). *Suppose for any sequence of actions $a_1, \ldots, a_T$ an online regression oracle $\mathcal{A}$ playing regressors $h_1, \ldots, h_T \in \mathcal{H}$ has regret guarantee*

$$R_T \geq \sum_{t=1}^{T} (\ell_t(c_t, a_t) - h_t(c_t, a_t))^2 - \min_{h \in \mathcal{H}} \sum_{t=1}^{T} (\ell_t(c_t, a_t) - h(c_t, a_t))^2 \tag{18}$$

*If all losses and regressors have range $[0, 1]$ and $\exists \ h \in \mathcal{H}$ s.t. $\mathbb{E}\ell_t(a) = h(c_t, a) + \alpha_t(c_t, a)$ for $|\alpha_t(a)| \leq \alpha$ then Algorithm 6 with learning rate $\eta = 2\sqrt{dT/(R_T + 2\alpha^2 T)}$ has expected regret w.r.t the the optimal policy $f : [a, b] \mapsto \mathbf{g}$ bounded as*

$$\mathbb{E} \sum_{t=1}^{T} \ell_t(\mathbf{g}_{[i_t]}) - \sum_{t=1}^{T} \ell_t(h(c_t)) \leq 2\sqrt{dT R_T} + 5\alpha T \sqrt{d} \tag{19}$$

**Algorithm 6:** SquareCB method for contextual bandits using an online regression oracle.

---

**Input:** loss sequence $\{\ell_t : \mathbf{g} \mapsto [0,1]\}_{t=1}^T$, context sequence $\{c_t\}_{t=1}^T$, learning rate $\eta > 0$,
online regression oracle $\mathcal{A}$
**for** $t = 1, \ldots, T$ **do**

   $\mathbf{s}_{[i]} \leftarrow \mathcal{A}(c_t, \mathbf{g}_{[i]}) \; \forall \, i \in [d]$                 `// compute oracle prediction`
   $i^* \leftarrow \arg\min_i \mathbf{s}_{[i]}$
   $\mathbf{p}_{[i]} \leftarrow \frac{1}{d + \eta(\mathbf{s}_{[i]} - \mathbf{s}_{[i^*]})} \; \forall \, i \neq i^*$         `// compute action probabilities`
   $\mathbf{p}_{[i^*]} \leftarrow 1 - \sum_{i \neq i^*} \mathbf{p}_{[i]}$
   sample $i_t \in [d]$ with probability $\mathbf{p}_{[i_t]}$      `// sample index of a grid point`
   $((c_t, \mathbf{g}_{[i_t]}), \ell_t(\mathbf{g}_{[i_t]})) \rightarrow \mathcal{A}$ `// pass context, action, and loss to oracle`

---

**Algorithm 7:** SquareCB method for Lipschitz contextual bandits using Follow-the-Leader.

---

**Input:** loss sequence $\{\ell_t : [a,b] \mapsto [0,K]\}_{t=1}^T$, context sequence $\{c_t \in [c, c+C]\}_{t=1}^T$, learning
     rate $\eta > 0$, action set $\mathbf{g} \in [a,b]^d$, featurizer $\mathbf{f} : [c, c+C] \mapsto \mathbb{R}^m$, normalizations $L, N > 0$
**for** $t = 1, \ldots, T$ **do**

   $\theta_i \leftarrow \underset{|\theta_{[0]}| \leq \frac{1}{N}, |\theta_{[j]}| \leq \frac{2CL}{KNj}}{\arg\min} \sum_{s \in [t-1]_i} \left( \langle \theta, \mathbf{f}(c_s) \rangle - \frac{\ell_t(\mathbf{g}_{[i_s]})}{KN} \right)^2 \; \forall \, i \in [d]$ `// update models`
   $\mathbf{s}_{[i]} \leftarrow \langle \theta_i, \mathbf{f}(c_t) \rangle \; \forall \, i \in [d]$                `// compute model predictions`
   $i^* \leftarrow \arg\min_i \mathbf{s}_{[i]}$
   $\mathbf{p}_{[i]} \leftarrow \frac{1}{d + \eta(\mathbf{s}_{[i]} - \mathbf{s}_{[i^*]})} \; \forall \, i \neq i^*$         `// compute action probabilities`
   $\mathbf{p}_{[i^*]} = 1 - \sum_{i \neq i^*} \mathbf{p}_{[i]}$
   sample $i_t \in [d]$ with probability $\mathbf{p}_{[i_t]}$ and play action $\mathbf{g}_{[i_t]}$

---

SquareCB requires an online regression oracle to implement, for which we will use the Follow-the-Leader scheme. It has the following guarantee for squared losses:

**Theorem C.3** (Corollary of Hazan et al. (2007, Theorem 5)). *Consider the follow-the-leader algorithm, which sequentially sees feature-target pairs $(\mathbf{x}_1, y_1), \cdots, (\mathbf{x}_T, y_T) \in \mathcal{X} \times [0,1]$ for some subset $\mathcal{X} \subset [0,1]^n$ and at each step sets $\theta_{t+1} = \arg\min_{\theta \in \Theta} \sum_{t=1}^T (\langle \mathbf{x}_t, \theta \rangle - y_t)^2$ for some subset $\Theta \subset \mathbb{R}^n$. This algorithm has regret*

$$\sum_{t=1}^T (\langle \mathbf{x}_t, \theta_t \rangle - y_t)^2 - \min_{\theta \in \Theta} (\langle \mathbf{x}_t, \theta \rangle - y_t)^2 \leq 4B^2 n \left( 1 + \log \frac{XDT}{2B} \right) \tag{20}$$

*for $D_\Theta$ the diameter $\max_{\theta, \theta'} \|\theta - \theta'\|_2$ of $\Theta$, $X = \max_{t \in [T]} \|\mathbf{x}_t\|_2$, and $B = \max_{t \in [T], \theta \in \Theta} |\langle \mathbf{x}_t, \theta \rangle|$.*

## C.2 REGRET OF CHEBCB

**Theorem C.4.** *Suppose $\mathbb{E}\ell_t(x)$ is an $L_x$-Lipschitz function of actions $x \in [a,b]$ and an $L_c$-Lipschitz function of contexts $c_t \in [c, c+C]$. Then Algorithm 7 run with learning rate $\eta = 2\sqrt{dT/(R_T + 2\alpha^2 T)}$ for $R_T$ and $\alpha$ as in Equations 22 and 23, respectively, action set $\mathbf{g}_{[i]} = a + (b-a)\frac{i - 1/2}{d}$, Chebyshev features $\mathbf{f}_{[j]}(c_t) = P_j(c_t)$, and normalizations $L = L_c$ and $N = 2 + \frac{4CL_c}{K}(1 + \log m)$ has regret w.r.t. any policy $f : [c, c+C] \mapsto [a,b]$ of*

$$\mathbb{E} \sum_{t=1}^T \ell_t(\mathbf{g}_{[i_t]}) - \ell_t(f(c_t)) = \tilde{\mathcal{O}} \left( L_c d\sqrt{mT} + \frac{L_c T \sqrt{d}}{m} + \frac{L_x T}{d} \right) \tag{21}$$

*Setting $d = \Theta(T^{2/11})$ and $m = \Theta(T^{3/11})$ yields a regret $\tilde{\mathcal{O}}(\max\{L_c, L_x\} T^{9/11})$.*

*Proof.* Observe that the above algorithm is equivalent to running Algorithm 6 with the follow-the-leader oracle over an $d(m+1)$-dimensional space $\Theta$ with diameter $\sqrt{\frac{d}{N^2}\left(1 + \frac{4C^2 L_c^2}{K^2}\sum_{j=1}^m \frac{1}{j^2}\right)} \leq \frac{\sqrt{dK^2 + 2dC^2 L_c^2 \pi^2/3}}{KN}$, features bounded by $\sqrt{1 + \sum_{j=1}^m P_j(c_t)} \leq \sqrt{m+1}$, and predictions bounded by $|\langle \mathbf{f}(c), \theta\rangle| \leq \|\theta\|_1 \|\mathbf{f}(c)\|_\infty \leq \frac{1}{N} + \frac{2CL_c}{KN}\sum_{j=1}^m \leq \frac{1}{2}$. Thus by Theorem C.3 the oracle has regret at most

$$R_T = d(m+1)\left(1 + \log\frac{T\sqrt{d(m+1)(K^2 + 2C^2 L_c^2 \pi^2/3)}}{KN}\right) \tag{22}$$

Note that, to ensure the regressors and losses have range in $[0, 1]$ we can define the former as $h(c, \mathbf{g}_{[i]}) = \langle \mathbf{f}(c), \theta_i\rangle + \frac{1}{2}$ and the latter as $\frac{\ell_t}{KN} + \frac{1}{2}$ and Algorithm 7 remains the same. Furthermore, the error of the regression approximation is then

$$\alpha = \frac{\pi + \frac{2}{\pi}\log(2m+1)}{2KN(m+1)}CL_c \tag{23}$$

We conclude by applying Theorem C.2, unnormalizing by multiplying the resulting regret by $KN$, and adding the approximation error $\frac{L_x(b-a)}{2d}$ due to the discretization of the action space. $\square$

# D SOR PRELIMINARIES

We will use the following notation:

- $\mathbf{M}_\omega = \mathbf{I}_n - (\mathbf{D}/\omega + \mathbf{L})^{-1}\mathbf{A}$ is the matrix of the first normal form (Hackbusch, 2016, 2.2.1)
- $\mathbf{W}_\omega = \mathbf{D}/\omega + \mathbf{L}$ is the matrix of the third normal form (Hackbusch, 2016, Section 2.2.3)
- $\mathbf{C}_\omega = \mathbf{I}_n - \mathbf{A}(\mathbf{D}/\omega + \mathbf{L})^{-1} = \mathbf{I}_n - \mathbf{A}\mathbf{W}_\omega^{-1} = \mathbf{A}\mathbf{M}_\omega\mathbf{A}^{-1}$ is the defect reduction matrix
- $\breve{\mathbf{M}}_\omega = \mathbf{I}_n - \frac{2-\omega}{\omega}(\mathbf{D}/\omega + \mathbf{L}^T)^{-1}\mathbf{D}(\mathbf{D}/\omega + \mathbf{L})^{-1}\mathbf{A}$ is the matrix of the first normal form for SSOR (Hackbusch, 2016, 2.2.1)
- $\breve{\mathbf{W}}_\omega = \frac{\omega}{2-\omega}(\mathbf{D}/\omega + \mathbf{L})\mathbf{D}^{-1}(\mathbf{D}/\omega + \mathbf{L}^T)$ is the matrix of the third normal form for SSOR (Hackbusch, 2016, Section 2.2.3)
- $\breve{\mathbf{C}}_\omega = \mathbf{I}_n - \frac{2-\omega}{\omega}\mathbf{A}(\mathbf{D}/\omega + \mathbf{L}^T)^{-1}\mathbf{D}(\mathbf{D}/\omega + \mathbf{L})^{-1} = \mathbf{I}_n - \mathbf{A}\breve{\mathbf{W}}_\omega^{-1} = \mathbf{A}\breve{\mathbf{M}}_\omega\mathbf{A}^{-1}$ is the defect reduction matrix for SSOR
- $\|\cdot\|_2$ denotes the Euclidean norm of a vector and the spectral norm of a matrix
- $\kappa(\mathbf{A}) = \|\mathbf{A}\|_2\|\mathbf{A}^{-1}\|_2$ denotes the condition number of a matrix $\mathbf{A} \succ 0$
- $\rho(\mathbf{X})$ denotes the spectral radius of a matrix $\mathbf{X}$
- $\|\mathbf{x}\|_{\mathbf{A}} = \|\mathbf{A}^{\frac{1}{2}}\mathbf{x}\|_2$ denotes the energy norm of a vector $\mathbf{x} \in \mathbb{R}^n$ associated with the matrix $\mathbf{A} \succ 0$
- $\|\mathbf{X}\|_{\mathbf{A}} = \|\mathbf{A}^{\frac{1}{2}}\mathbf{X}\mathbf{A}^{-\frac{1}{2}}\|_2$ denotes the energy norm of a matrix $\mathbf{X} \in \mathbb{R}^{n\times n}$ associated with the matrix $\mathbf{A} \succ 0$

We further derive bounds on the number of iterations for SOR and SSOR using the following energy norm estimate:

**Theorem D.1** (Corollary of Hackbusch (2016, Theorem 3.44 & Corollary 3.45)). *If $\mathbf{A} \succ 0$ and $\omega \in (0,2)$ then $\|\mathbf{M}_\omega\|_{\mathbf{A}}^2 \leq 1 - \frac{\frac{2-\omega}{\omega}\gamma}{\left(\frac{2-\omega}{2\omega}\right)^2 + \frac{\gamma}{\omega} + \rho(\mathbf{D}^{-1}\mathbf{L}\mathbf{D}^{-1}\mathbf{L}^T) - \frac{1}{4}}$, where $\gamma = 1 - \rho(\mathbf{D}^{-1}(\mathbf{L} + \mathbf{L}^T))$.*

**Corollary D.1.** *Let $K_\omega$ be the maximum number of iterations that SOR needs to reach error $\varepsilon > 0$. Then for any $\omega \in (0,2)$ we have $K_\omega \leq 1 + \frac{-\log\frac{\varepsilon}{2\sqrt{\kappa(A)}}}{-\log \nu_\omega(\mathbf{A})}$, where $\nu_\omega(\mathbf{A})$ is the square root of the upper bound in Theorem D.1.*

*Proof.* By Hackbusch (2016, Equations 2.22c & B.28b) we have at each iteration $k$ of Algorithm 1 that

$$\frac{\|\mathbf{r}_k\|_2}{\|\mathbf{r}_0\|_2} \leq \frac{2}{\|\mathbf{r}_0\|_2}\|\mathbf{A}^{\frac{1}{2}}\|_2\|\mathbf{M}_\omega^k\|_{\mathbf{A}}\|\mathbf{A}^{-\frac{1}{2}}\mathbf{r}_0\|_2 \leq 2\sqrt{\kappa(\mathbf{A})}\|\mathbf{M}_\omega\|_{\mathbf{A}}^k \tag{24}$$

Setting the r.h.s. equal to $\varepsilon$ and solving for $k$ yields the result. $\qquad\square$

**Corollary D.2.** *Let $\breve{K}_\omega$ be the maximum number of iterations that SSOR (Algorithm 8) needs to reach (absolute) error $\varepsilon > 0$. Then for any $\omega \in (0,2)$ we have $\breve{K}_\omega \leq 1 + \frac{-\log\frac{\varepsilon}{2\|\mathbf{b}\|_2\sqrt{\kappa(A)}}}{-2\log \nu_\omega(\mathbf{A})}$, where $\nu_\omega(\mathbf{A})$ is the square root of the upper bound in Theorem D.1.*

*Proof.* By Hackbusch (2016, Equations 2.22c & B.28b) we have at each iteration $k$ of Algorithm 1 that

$$\|\mathbf{r}_k\|_2 \leq 2\|\mathbf{A}^{\frac{1}{2}}\|_2\|\breve{\mathbf{M}}_\omega^k\|_{\mathbf{A}}\|\mathbf{A}^{-\frac{1}{2}}\mathbf{r}_0\|_2 \leq 2\|\mathbf{b}\|_2\sqrt{\kappa(\mathbf{A})}\|\mathbf{M}_\omega\|_{\mathbf{A}}^{2k} \tag{25}$$

Setting the r.h.s. equal to $\varepsilon$ and solving for $k$ yields the result. $\qquad\square$

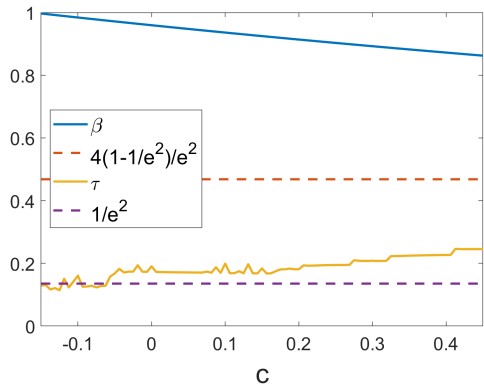

Figure 3: Values of $\tau$ and $\beta$ for $\mathbf{A} + c\mathbf{I}_n$ for different $c$.

# E    NEAR-ASYMPTOTIC PROOFS

## E.1    PROOF OF LEMMA 2.1 AND ASSOCIATED DISCUSSION

*Proof.* For the first claim, suppose $l = \min\limits_{\|\mathbf{C}_\omega^k \mathbf{b}\|_2 < \varepsilon \|\mathbf{b}\|_2} k > U(\omega)$. Then

$$\varepsilon \leq \frac{\|\mathbf{C}_\omega^{l-1}\mathbf{b}\|_2}{\|\mathbf{b}\|_2} \leq \frac{\|\mathbf{C}_\omega^{l-1}\|_2 \|\mathbf{b}\|_2}{\|\mathbf{b}\|_2} \leq (\rho(\mathbf{C}_\omega) + \tau(1 - \rho(\mathbf{C}_\omega)))^{l-1} < \varepsilon \tag{26}$$

so by contradiction we must have $\min\limits_{\|\mathbf{C}_\omega^k \mathbf{b}\|_2 < \varepsilon \|\mathbf{b}\|_2} k \leq U(\omega)$. Now note that by Hackbusch (2016,

Theorem 4.27) and similarity of $\mathbf{C}_\omega$ and $\mathbf{M}_\omega$ we have that $\rho(\mathbf{C}_\omega) = \frac{1}{4}\left(\omega\beta + \sqrt{\omega^2\beta^2 - 4(\omega - 1)}\right)^2$

for $\omega < \omega^* = 1 + \left(\frac{\beta}{1+\sqrt{1-\beta^2}}\right)^2$ and $\omega - 1$ otherwise. Therefore on $\omega < \omega^*$ we have $\rho(\mathbf{C}_\omega) \leq$

$\rho(\mathbf{C}_1) = \beta^2$ and on $\omega \geq \omega^*$ we have $\rho(\mathbf{C}_\omega) \leq \omega_{\max} - 1$. This concludes the second part of the
first claim. The first part of the second claim follows because $\rho(\mathbf{C}_\omega)$ is decreasing on $\omega < \omega^*$. For
the second part, we compute the derivative $|\partial_\omega U(\omega)| = \frac{(\tau - 1)\log\varepsilon}{(\tau + (1-\tau)(\omega - 1))\log^2(\tau + (1-\tau)(\omega - 1))}$. Since
$\tau + (1 - \tau)(\omega - 1) \geq \frac{1}{e^2}$ by assumption—either by nonnegativity of $\omega - 1$ if $\tau \geq \frac{1}{e^2}$ or because
otherwise $\beta^2 \geq \frac{4}{e^2}(1 - \frac{1}{e^2})$ implies $\tau + (1 - \tau)(\omega - 1) \geq (1 - \frac{1}{e^2})\left(\frac{\beta}{1+\sqrt{1-\beta^2}}\right)^2 \geq \frac{1}{e^2}$—the
derivative is increasing in $\omega$ and so is at most $\frac{-(1-\tau)\log\varepsilon}{\alpha\log^2\alpha}$. $\square$

Note that the fourth item's restriction on $\tau$ and $\beta$ does not really restrict the matrices our analysis
is applicable to, as we can always re-define $\tau$ in Assumption 2.1 to be at least $1/e^2$. Our analysis
does not strongly depend on this restriction; it is largely done for simplicity and because it does not
exclude too many settings of interest. In-particular, we find for $\varepsilon \geq 10^{-8}$ that $\tau$ is typically indeed
larger than $\frac{1}{e^2}$, and furthermore $\tau$ is likely quite high whenever $\beta$ is small, as it suggests the matrix is
near-diagonal and so $\omega$ near one will converge extremely quickly (c.f. Figure 3).

## E.2    PROOF OF THEOREM 2.1

*Proof.* The first bound follows from Theorem B.3 by noting that Lemma 2.1 implies that the functions
$U_t - 1$ are $\left(\frac{-(1-\tau_t)\log\varepsilon}{\alpha_t\log^2\alpha_t}, \omega_{\max}\right)$-semi-Lipschitz over $[1, \omega_{\max}]$ and the functions $\texttt{SOR}_t - 1 \leq U_t - 1$
are $\frac{-\log\varepsilon}{-\log\alpha_t}$-bounded. To extend the comparator domain to $(0, \omega_{\max}]$, note that Lemma 2.1.2 implies
that all $U_t$ are decreasing on $\omega \in (0, 1)$. To extend the comparator domain again in the second bound,
note that the setting of $\omega_{\max}$ implies that the minimizer $1 + \beta_t^2/(1 + \sqrt{1 - \beta_t^2})$ of each $U_t$ is at most
$\omega_{\max}$, and so all functions $U_t$ are increasing on $\omega \in (\omega_{\max}, 2)$. $\square$

### E.3 Approximating the optimal policy

**Lemma E.1.** *Define* $\mathbf{A}(c) = \mathbf{A} + c\mathbf{I}_n$ *for all* $c \in [c_{\min}, \infty)$, *where* $c_{\min} > -\lambda_{\min}(\mathbf{A})$. *Then* $\omega^*(c) = 1 + \left( \frac{\beta_c}{\sqrt{1-\beta_c^2}+1} \right)^2$ *is* $\frac{6\beta_{\max}(1+\beta_{\max})/\sqrt{1-\beta_{\max}^2}}{\left( \sqrt{1-\beta_{\max}^2}+1 \right)^2} \left( \frac{\lambda_{\min}(\mathbf{D})+c_{\min}+1}{\lambda_{\min}(\mathbf{D})+c_{\min}} \right)^2$-*Lipschitz, where* $\beta_{\max} = \max_c \beta_c$ *is the maximum over* $\beta_c = \rho(\mathbf{I}_n - (\mathbf{D}+c\mathbf{I}_n)^{-1}(\mathbf{A}+c\mathbf{I}_n))$.

*Proof.* We first compute

$$
\begin{aligned}
\partial_c \beta_c &= \partial_c \rho(\mathbf{I}_n - (\mathbf{D}+c\mathbf{I}_n)^{-1}(\mathbf{A}+c\mathbf{I}_n)) \\
&= \partial_c \lambda_{\max}(\mathbf{I}_n - (\mathbf{D}+c\mathbf{I}_n)^{-\frac{1}{2}}(\mathbf{A}+c\mathbf{I}_n)(\mathbf{D}+c\mathbf{I}_n)^{-\frac{1}{2}}) \\
&= \mathbf{v}_1^T \partial_c((\mathbf{I}_n - (\mathbf{D}+c\mathbf{I}_n)^{-\frac{1}{2}}(\mathbf{A}+c\mathbf{I}_n)(\mathbf{D}+c\mathbf{I}_n)^{-\frac{1}{2}}))\mathbf{v}_1 \\
&= -\mathbf{v}_1^T (\partial_c((\mathbf{D}+c\mathbf{I}_n)^{-\frac{1}{2}})(\mathbf{A}+c\mathbf{I}_n)(\mathbf{D}+c\mathbf{I}_n)^{-\frac{1}{2}} \\
&\qquad + (\mathbf{D}+c\mathbf{I}_n)^{-\frac{1}{2}}\partial_c(\mathbf{A}+c\mathbf{I}_n)(\mathbf{D}+c\mathbf{I}_n)^{-\frac{1}{2}} \\
&\qquad + (\mathbf{D}+c\mathbf{I}_n)^{-\frac{1}{2}}(\mathbf{A}+c\mathbf{I}_n)\partial_c((\mathbf{D}+c\mathbf{I}_n)^{-\frac{1}{2}}))\mathbf{v}_1 \\
&= \frac{1}{2}\mathbf{v}_1^T((\mathbf{D}+c\mathbf{I}_n)^{-\frac{3}{2}}(\mathbf{A}+c\mathbf{I}_n)(\mathbf{D}+c\mathbf{I}_n)^{-\frac{1}{2}} - 2c(\mathbf{D}+c\mathbf{I}_n)^{-1} \\
&\qquad + (\mathbf{D}+c\mathbf{I}_n)^{-\frac{1}{2}}(\mathbf{A}+c\mathbf{I}_n)(\mathbf{D}+c\mathbf{I}_n)^{-\frac{3}{2}})\mathbf{v}_1 \\
&= \frac{1}{2}\mathbf{v}_1^T(\mathbf{I}_n + (\mathbf{D}+c\mathbf{I}_n)^{-1})(\mathbf{D}+c\mathbf{I}_n)^{-\frac{1}{2}}(\mathbf{A}+c\mathbf{I})(\mathbf{D}+c\mathbf{I}_n)^{-\frac{1}{2}}(\mathbf{I}_n + (\mathbf{D}+c\mathbf{I}_n)^{-1})\mathbf{v}_1 \\
&\qquad - \frac{1}{2}\mathbf{v}_1^T(\mathbf{D}+c\mathbf{I}_n)^{-\frac{1}{2}}(\mathbf{A}+c\mathbf{I}_n)(\mathbf{D}+c\mathbf{I}_n)^{-\frac{1}{2}}\mathbf{v}_1 \\
&\qquad - \frac{1}{2}\mathbf{v}_1^T(\mathbf{D}+c\mathbf{I}_n)^{-\frac{3}{2}}(\mathbf{A}+c\mathbf{I}_n)(\mathbf{D}+c\mathbf{I}_n)^{-\frac{3}{2}}\mathbf{v}_1 - c\mathbf{v}_1^T(\mathbf{D}+c\mathbf{I}_n)^{-1}\mathbf{v}_1 \\
&= \frac{1}{2}\mathbf{v}_1^T(\mathbf{I}_n + (\mathbf{D}+c\mathbf{I}_n)^{-1})(\mathbf{D}+c\mathbf{I}_n)^{-\frac{1}{2}}(\mathbf{A}+c\mathbf{I})(\mathbf{D}+c\mathbf{I}_n)^{-\frac{1}{2}}(\mathbf{I}_n + (\mathbf{D}+c\mathbf{I}_n)^{-1})\mathbf{v}_1 \\
&\qquad - \frac{1}{2}\mathbf{v}_1^T(\mathbf{D}+c\mathbf{I}_n)^{-\frac{1}{2}}(\mathbf{A}+3c\mathbf{I}_n)(\mathbf{D}+c\mathbf{I}_n)^{-\frac{1}{2}}\mathbf{v}_1 \\
&\qquad - \frac{1}{2}\mathbf{v}_1^T(\mathbf{D}+c\mathbf{I}_n)^{-\frac{3}{2}}(\mathbf{A}+c\mathbf{I}_n)(\mathbf{D}+c\mathbf{I}_n)^{-\frac{3}{2}}\mathbf{v}_1
\end{aligned}
\tag{27}
$$

The first component is positive and the matrix has eigenvalues bounded by $\left(1 + \frac{1}{\lambda_{\min}(\mathbf{D})+c}\right)^2 \frac{1+\beta_c}{2}$, while the last term is negative and the matrix has If $c \geq 0$ the positive component has spectral radius at most $\frac{1+\beta_c}{2}\left(1 + \frac{1}{(\lambda_{\min}(\mathbf{D})+c)^2}\right)$. If the middle term is negative, subtracting $2\mathbf{A}$ from the middle matrix shows that its magnitude is bounded by $\frac{3}{2}(1+\beta_c)$. If the middle term is positive—which can only happen for negative $c$—its magnitude is bounded by $\frac{-3c/2}{\lambda_{\min}(\mathbf{D})+c} \leq \frac{3\lambda_{\min}(\mathbf{D})/2}{\lambda_{\min}(\mathbf{D})+c}$. Combining all terms yields a bound of $3\left(\frac{\lambda_{\min}(\mathbf{D})+c+1}{\lambda_{\min}(\mathbf{D})+c}\right)^2(1+\beta_c)$. We then have that

$$
|\partial_c\omega^*(c)| = \frac{2\beta_c/\sqrt{1-\beta_c^2}}{\left(\sqrt{1-\beta_c^2}+1\right)^2}|\partial_c\beta_c| = \frac{6\beta_c(1+\beta_c)/\sqrt{1-\beta_c^2}}{\left(\sqrt{1-\beta_c^2}+1\right)^2}\left(\frac{\lambda_{\min}(\mathbf{D})+c+1}{\lambda_{\min}(\mathbf{D})+c}\right)^2
\tag{28}
$$

The result follows because $\frac{2x(1+x)/\sqrt{1-x^2}}{\left(\sqrt{1-x^2}+1\right)^2}$ increases monotonically on $x \in [0,1)$ and the bound itself decreases monotonically in $c$ $\qquad\square$

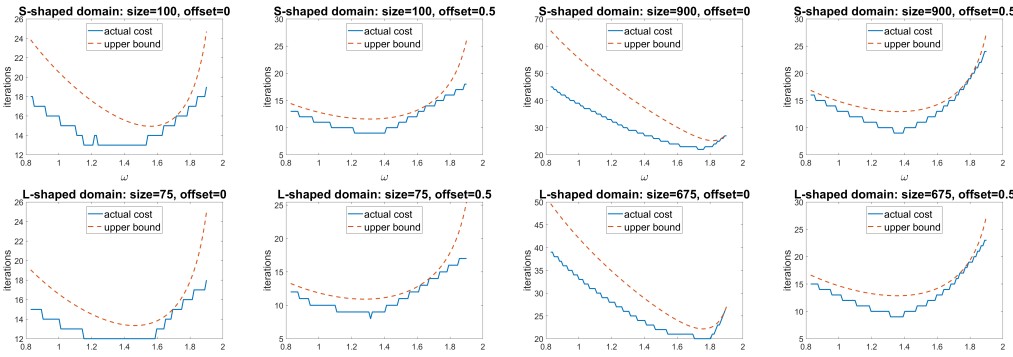

Figure 4: Comparison of actual cost of running SSOR-preconditioned CG and the upper bounds computed in Section E.4 as functions of the tuning parameter $\omega \in [2\sqrt{2}-2, 1.9]$ on various domains.

**Theorem E.1.** *Suppose $c_t \in [c_{\min}, c_{\min} + C] \ \forall \, t \in [T]$, where $c_{\min} > -\lambda_{\min}(\mathbf{A})$, and define $\beta_{\max} = \max_t \beta_t$. Then if we run Algorithm 5 with losses $(\text{SOR}_t(\cdot) - 1)/K$ normalized by $K \geq \frac{-\log \varepsilon}{-\log \alpha_{\max}}$ for $\alpha_{\max} = \max_t \alpha_t$, action set $\mathbf{g}_{[i]} = 1 + (\omega_{\max} - 1)\frac{i}{d}$ for $\omega_{\max} \geq 1 + \left(\frac{\beta_{\max}}{1+\sqrt{1-\beta_{\max}^2}}\right)^2$, and context discretization $\mathbf{h}_{[j]} = c_{\min} + \frac{C}{m}\left(j - \frac{1}{2}\right)$, then the number of iterations will be bounded in expectation as*

$$\mathbb{E}\sum_{t=1}^{T} \text{SOR}_t(\omega_t) \leq m + 4K\sqrt{dmT} + \left(\frac{CL^*/m}{\omega_{\max} - 1} + \frac{1}{d}\right)\sum_{t=1}^{T}\frac{-\log \varepsilon}{\log^2 \alpha_t} + \sum_{t=1}^{T}U_t(\omega^*(c_t)) \qquad (29)$$

*where $L^*$ is the Lipschitz constant from Lemma E.1. Setting $\omega_{\max} = 1 + \left(\frac{\beta_{\max}^2}{1+\sqrt{1-\beta_{\max}^2}}\right)^2$, $K = \frac{-\log \varepsilon}{-\log \alpha_{\max}}$, $d = \sqrt[4]{\frac{\bar{\gamma}^2 T \log^2 \alpha_{\max}}{24CL}}$, and $m = \sqrt[4]{54C^3L^3\bar{\gamma}^2T\log^2\alpha_{\max}}$ where $\bar{\gamma} = \frac{1}{T}\sum_{t=1}^{T}\frac{1}{\log^2 \alpha_t}$ and $\tilde{L} = \left(\frac{\lambda_{\min}(\mathbf{D})+c_{\min}+1}{\lambda_{\min}(\mathbf{D})+c_{\min}}\right)^2 \frac{1+\beta_{\max}}{\beta_{\max}\sqrt{1-\beta_{\max}^2}}$, yields*

$$E\sum_{t=1}^{T}\text{SOR}_t(\omega_t) \leq \sqrt[4]{54C^3L^3\bar{\gamma}^2T\log^2\alpha_{\max}} + 4\log\frac{1}{\varepsilon}\sqrt[4]{\frac{24CL\bar{\gamma}T^3}{\log^2\alpha_{\max}}} + \sum_{t=1}^{T}U_t(\omega^*(c_t))$$

$$\leq \sqrt[4]{\frac{54C^3L^3T}{\log^2\alpha_{\max}}} + \frac{4\log\frac{1}{\varepsilon}}{\log\frac{1}{\alpha_{\max}}}\sqrt[4]{24CLT^3} + \sum_{t=1}^{T}U_t(\omega^*(c_t)) \qquad (30)$$

*Proof.* The bound follows from Theorem B.4 by noting that Lemma 2.1 implies that the functions $U_t - 1$ are $\left(\frac{-(1-\tau_t)\log \varepsilon}{\alpha_t \log^2 \alpha_t}, \omega_{\max}\right)$-semi-Lipschitz over $[1, \omega_{\max}]$ and the functions $\text{SOR}_t - 1 \leq U_t - 1$ are $\frac{-\log \varepsilon}{-\log \alpha_t}$-bounded. Note that for the choice of $\omega_{\max}$ we the interval $[1, \omega_{\max}]$ contains the range of the optimal policy $\omega^*$, and further by Lemma E.1 it is $L^*$-Lipschitz over $[c_{\min}, c_{\min} + C]$. □

### E.4 EXTENSION TO PRECONDITIONED CG

While CG is an iterative algorithm, for simplicity we define it as the solution to a minimization problem in the Krylov subspace:

**Definition E.1.** $CG(\mathbf{A}, \mathbf{b}, \omega) = \min_{\|\mathbf{A}\mathbf{x}_k - \mathbf{b}\|_2 \leq \varepsilon} k$ *for* $\mathbf{x}_k = \underset{\mathbf{x} = \mathbf{P}_k(\check{\mathbf{W}}_\omega^{-1}\mathbf{A})\check{\mathbf{W}}_\omega^{-1}\mathbf{b}}{\arg\min} \|\mathbf{x} - \mathbf{A}^{-1}\mathbf{b}\|_{\mathbf{A}}$ *where the minimum is taken over all degree $k$ polynomials* $\mathbf{P}_k : \mathbb{R}^{n \times n} \mapsto \mathbb{R}^{n \times n}$.

**Lemma E.2.** *Let* $\mathbf{A}$ *be a positive-definite matrix and* $\mathbf{b} \in \mathbb{R}^n$ *any vector. Define*

$$U^{CG}(\omega) = 1 + \frac{\tau \log\left(\frac{\sqrt{\kappa(\mathbf{A})}}{\varepsilon} + \sqrt{\frac{\kappa(\mathbf{A})}{\varepsilon^2} - 1}\right)}{-\log\left(1 - \frac{4}{2 + \sqrt{\frac{4}{2-\omega} + \frac{\mu(2-\omega)}{\omega} + \frac{4\nu\omega}{2-\omega}}}\right)} \tag{31}$$

*for* $\mu = \lambda_{\max}(\mathbf{D}\mathbf{A}^{-1}) \geq 1$, $\nu = \lambda_{\max}((\mathbf{L}\mathbf{D}^{-1}\mathbf{L}^T - \mathbf{D}/4)\mathbf{A}^{-1}) \in [-1/4, 0]$, *and* $\tau$ *the smallest constant (depending on* $\mathbf{A}$ *and* $\mathbf{b}$) *s.t.* $U^{CG} \geq CG(\mathbf{A}, \mathbf{b}, \cdot)$. *Then the following holds*

1. $\tau \in (0, 1]$

2. *if* $\mu > 1$ *then* $U^{CG}$ *is minimized at* $\omega^* = \frac{2}{1 + \sqrt{\frac{2}{\mu}(1 + 2\nu)}}$ *and monotonically increases away from* $\omega^*$ *in both directions*

3. $U^{CG}$ *is* $\left(\frac{\mu + 4\nu + 4}{4\mu\nu + 2\mu - 1} \tau \sqrt{\mu\sqrt{2}}, 2\sqrt{2} - 2\right)$-*semi-Lipschitz on* $[2\sqrt{2} - 2, 2)$

4. *if* $\mu \leq \mu_{\max}$ *then* $U^{CG} \leq 1 + \frac{\tau \log(\frac{2}{\varepsilon}\sqrt{\kappa(\mathbf{A})})}{-\log\left(1 - \frac{2}{1 + \sqrt{\gamma}}\right)}$ *on* $[2\sqrt{2} - 2, \frac{2}{1 + 1/\sqrt{\mu_{\max}}}]$, *where* $\gamma \leq \frac{7 + 3\mu_{\max}}{8}$.

*Proof.* By Hackbusch (2016, Theorem 10.17) we have that the $k$th residual of SSOR-preconditioned CG satisfies

$$\|\mathbf{r}_k(\omega)\|_2 = \|\mathbf{b} - \mathbf{A}\mathbf{x}_k\|_2 \leq \sqrt{\|\mathbf{A}\|}\|\mathbf{A}^{-1}\mathbf{b} - \mathbf{x}_k\|_{\mathbf{A}} \leq \sqrt{\|\mathbf{A}\|}\frac{2x^k}{1 + x^{2k}}\|\mathbf{A}^{-1}\mathbf{b} - \mathbf{x}_0\|_{\mathbf{A}}$$
$$\leq \frac{2\sqrt{\kappa(\mathbf{A})}x^k}{1 + x^{2k}}\|\mathbf{r}_0\|_2 \tag{32}$$

for $x = \frac{\sqrt{\kappa(\breve{\mathbf{W}}_\omega^{-1}\mathbf{A})} - 1}{\sqrt{\kappa(\breve{\mathbf{W}}_\omega^{-1}\mathbf{A})} + 1} = 1 - \frac{2}{\sqrt{\kappa(\breve{\mathbf{W}}_\omega^{-1}\mathbf{A})} + 1}$. By Axelsson (1994, Theorem 7.17) we have

$$\kappa(\breve{\mathbf{W}}_\omega^{-1}\mathbf{A}) \leq \frac{1 + \frac{\mu}{4\omega}(2 - \omega)^2 + \omega\nu}{2 - \omega} \tag{33}$$

Combining the two inequalities above yields the first result. For the second, we compute the derivative w.r.t. $\omega$:

$$\frac{\partial_\omega U^{CG}}{\tau} = \frac{8(2\nu + 1)\omega^2 - 4\mu(2 - \omega)^2}{(2 - \omega)\omega\sqrt{\frac{4}{2-\omega} + \frac{\mu(2-\omega)}{\omega} + \frac{4\nu\omega}{2-\omega}}(\mu(2 - \omega)^2 + 4\omega(\nu\omega + \omega - 1))} \tag{34}$$

Since $\nu \in [-1/4, 0]$ and $\mu > 1$, we have that $\mu(2 - \omega)^2 + 4\omega(\nu\omega + \omega - 1) \geq (2 - \omega)^2 + 3\omega^2 - 4\omega$, which is nonnegative. Therefore the derivative only switches signs once, at the zero of specified in the second result. The monotonic increase property follows by positivity of the numerator on $\omega > \omega^*$. The third property follows by noting that since $U^{CG}$ is increasing on $\omega > \omega^*$ we only needs to consider $\omega \in [2\sqrt{2} - 2, \omega^*]$, where the numerator of the derivative is negative; here we have

$$\frac{|\partial_\omega U^{CG}|}{\tau} \leq \frac{4\mu(2 - \omega)}{\omega\sqrt{\frac{4}{2-\omega} + \frac{\mu(2-\omega)}{\omega} + \frac{4\nu\omega}{2-\omega}}(\mu(2 - \omega)^2 + 4\omega(\nu\omega + \omega - 1))} \leq \frac{\mu + 4\nu + 4}{4\mu\nu + 2\mu - 1}\sqrt{\mu\sqrt{2}}$$
$$\tag{35}$$

where we have used $\mu(2 - \omega)^2 + 4\omega(\nu\omega + \omega - 1) \geq \frac{16\mu\nu + 8\mu - 4}{\mu + 4\nu + 4}$ and $\omega \geq 2\sqrt{2} - 2$. For the last result we use the fact that $\kappa(\breve{\mathbf{W}}_\omega^{-1}\mathbf{A})$ is maximal at the endpoints of the interval and evaluate it on those endpoints to bound $\gamma \leq \frac{1}{2} + \frac{\max\{(\mu_{\max} + 1)\sqrt{2}, 3\sqrt{\mu_{\max}}\}}{4} \leq \frac{1}{2} + \frac{3(\mu_{\max} + 1)}{8}$. $\qquad\square$

We plot the bounds from Lemma E.2 in Figure 4. Note that $\tau \in (0, 1]$ is an instance-dependent parameter that is defined to effectively scale down the function as much as possible while still being an upper bound on the cost; it thus allows us to exploit the shape of the upper bound without having it be too loose. This is useful since upper bounds for CG are known to be rather pessimistic, and we are able to do this because our learning algorithms do not directly access the upper bound anyway. Empirically, we find $\tau$ to often be around $3/4$ or larger.

### E.4.1 PROOF OF THEOREM 2.3

*Proof.* By Lemma E.2 the functions $U_t - 1 \geq \text{CG}_t - 1$ are $\left(\frac{\mu_t + 4\nu_t + 4}{4\mu_t\nu_t + 2\mu_t - 1}\sqrt{\mu_t\sqrt{2}}, 2\sqrt{2} - 2\right)$-semi-Lipschitz and $\frac{\log(\frac{2}{\varepsilon}\sqrt{\kappa_{\max}})}{\log\frac{\sqrt{6\mu_{\max}+14}+4}{\sqrt{6\mu_{\max}+14}-4}}$-bounded on $[2\sqrt{2} - 2, \frac{2}{1+1/\sqrt{\mu_{\max}}}]$; note that by the assumption on $\min_t \mu_t$ and the fact that $\nu_t \geq 1/4$ the semi-Lipschitz constant is $\mathcal{O}(\sqrt{\mu_t})$. Therefore the desired regret w.r.t. any $\omega \in [2\sqrt{2} - 2, \frac{2}{1+1/\sqrt{\mu_{\max}}}]$ follows, and extends to the rest of the interval because Lemma E.2.2 also implies all functions $U_t$ are increasing away from this interval. $\square$

# F  SEMI-STOCHASTIC PROOFS

## F.1  REGULARITY OF THE CRITERION

**Lemma F.1.** $\|\check{\mathbf{C}}_\omega^k \mathbf{b}\|_2$ *is* $\rho(\check{\mathbf{C}}_\omega)^{k-1}\|\mathbf{b}\|_2 k \sqrt{\kappa(\mathbf{A})} \left( \frac{1}{2-\omega_{\max}} + 2\rho(\mathbf{D}\mathbf{A}^{-1}) \right)$*-Lipschitz w.r.t.* $\omega \in \Omega$.

*Proof.* Taking the derivative, we have that

$$
\begin{aligned}
|\partial_\omega \|\check{\mathbf{C}}_\omega^k \mathbf{b}\|_2| &= \frac{|\partial_\omega[(\check{\mathbf{C}}_\omega^k \mathbf{b})^T \check{\mathbf{C}}_\omega^k \mathbf{b}]|}{\|\check{\mathbf{C}}_\omega^k \mathbf{b}\|_2} \\
&= \frac{\left| (\check{\mathbf{C}}_\omega^k \mathbf{b})^T \sum_{i=1}^k \left[ \check{\mathbf{C}}_\omega^{i-1}(\partial_\omega \check{\mathbf{C}}_\omega)\check{\mathbf{C}}_\omega^{k-i}\mathbf{b} \right] \right|}{\|\check{\mathbf{C}}_\omega^k \mathbf{b}\|_2} \\
&\leq \left\| \sum_{i=1}^k \check{\mathbf{C}}_\omega^{i-1}(\partial_\omega \check{\mathbf{C}}_\omega)\check{\mathbf{C}}_\omega^{k-i}\mathbf{b} \right\|_2 \\
&\leq \sum_{i=1}^k \left\| \check{\mathbf{C}}_\omega^{i-1}(\partial_\omega \check{\mathbf{C}}_\omega)\check{\mathbf{C}}_\omega^{k-i}\mathbf{b} \right\|_2 \\
&= \sum_{i=1}^k \left\| \mathbf{A}^{\frac{1}{2}}\mathbf{A}^{-\frac{1}{2}}\check{\mathbf{C}}_\omega^{i-1}\mathbf{A}^{\frac{1}{2}}\mathbf{A}^{-\frac{1}{2}}(\partial_\omega \check{\mathbf{C}}_\omega)\mathbf{A}^{\frac{1}{2}}\mathbf{A}^{-\frac{1}{2}}\check{\mathbf{C}}_\omega^{k-i}\mathbf{A}^{\frac{1}{2}}\mathbf{A}^{-\frac{1}{2}}\mathbf{b} \right\|_2 \\
&\leq \|\mathbf{b}\|_2 \sqrt{\kappa(\mathbf{A})} \sum_{i=1}^k \|(\mathbf{A}^{-\frac{1}{2}}\check{\mathbf{C}}_\omega \mathbf{A}^{\frac{1}{2}})^{i-1}\|_2 \|\mathbf{A}^{-\frac{1}{2}}(\partial_\omega \check{\mathbf{C}}_\omega)\mathbf{A}^{\frac{1}{2}}\|_2 \|(\mathbf{A}^{-\frac{1}{2}}\check{\mathbf{C}}_\omega \mathbf{A}^{\frac{1}{2}})^{k-i}\|_2 \\
&= \rho(\check{\mathbf{C}}_\omega)^{k-1}\|\mathbf{b}\|_2 k \|\mathbf{A}^{-\frac{1}{2}}(\partial_\omega \check{\mathbf{C}}_\omega)\mathbf{A}^{\frac{1}{2}}\|_2 \sqrt{\kappa(\mathbf{A})}
\end{aligned}
\tag{36}
$$

where the first inequality is due to Cauchy-Schwartz, the second is the triangle inequality, and the third is due to the sub-multiplicativity of the norm. The last line follows by symmetry of $\mathbf{A}^{-\frac{1}{2}}\check{\mathbf{C}}_\omega \mathbf{A}^{\frac{1}{2}}$, which implies that the spectral norm of any of power equals that power of its spectral radius, which by similarity is also the spectral radius of $\check{\mathbf{C}}_\omega$. Next we use a matrix calculus tool (Laue et al., 2018) to compute

$$
\begin{aligned}
\partial_\omega \check{\mathbf{C}}_\omega &= \left( \frac{1}{\omega} + \frac{2-\omega}{\omega^2} \right) \mathbf{A}(\mathbf{D}/\omega + \mathbf{L}^T)^{-1}\mathbf{D}(\mathbf{D}/\omega + \mathbf{L})^{-1} \\
&\quad - \frac{2-\omega}{\omega^3}\mathbf{A}(\mathbf{D}/\omega + \mathbf{L}^T)^{-1}\mathbf{D}(\mathbf{D}/\omega + \mathbf{L}^T)^{-1}\mathbf{D}(\mathbf{D}/\omega + \mathbf{L})^{-1} \\
&\quad + \frac{2-\omega}{\omega^3}\mathbf{A}(\mathbf{D}/\omega + \mathbf{L}^T)^{-1}\mathbf{D}(\mathbf{D}/\omega + \mathbf{L})^{-1}\mathbf{D}(\mathbf{D}/\omega + \mathbf{L})^{-1} \\
&= \left( \frac{1}{2-\omega} + \frac{1}{\omega} \right) \mathbf{A}\check{\mathbf{W}}_\omega^{-1} - \frac{1}{\omega^2}\mathbf{A}(\mathbf{D}/\omega + \mathbf{L}^T)^{-1}\mathbf{D}\check{\mathbf{W}}_\omega^{-1} - \frac{1}{\omega^2}\mathbf{A}\check{\mathbf{W}}_\omega^{-1}\mathbf{D}(\mathbf{D}/\omega + \mathbf{L})^{-1}
\end{aligned}
\tag{37}
$$

so since $\|\mathbf{A}^{\frac{1}{2}}\check{\mathbf{W}}_\omega^{-1}\mathbf{A}^{\frac{1}{2}}\|_2 = \|\mathbf{I}_n - \mathbf{A}^{-\frac{1}{2}}\check{\mathbf{C}}_\omega \mathbf{A}^{\frac{1}{2}}\|_2 \leq 1 + \rho(\mathbf{A}^{-\frac{1}{2}}\check{\mathbf{C}}_\omega \mathbf{A}^{\frac{1}{2}}) \leq 2$ and

$$
\begin{aligned}
\|\mathbf{A}^{\frac{1}{2}}(\mathbf{D}/\omega + \mathbf{L}^T)^{-1}\mathbf{D}\check{\mathbf{W}}_\omega^{-1}\mathbf{A}^{\frac{1}{2}}\|_2 &= \|\mathbf{A}^{\frac{1}{2}}\check{\mathbf{W}}_\omega^{-1}\mathbf{D}(\mathbf{D}/\omega + \mathbf{L})^{-1}\mathbf{A}^{\frac{1}{2}}\|_2 \\
&= \|\check{\mathbf{W}}_\omega^{-1}\mathbf{D}\mathbf{W}_\omega^{-1}\mathbf{A}\|_{\mathbf{A}} \\
&\leq \|\check{\mathbf{W}}_\omega^{-1}\mathbf{D}\|_{\mathbf{A}}\|\mathbf{I}_n - \mathbf{M}_\omega\|_{\mathbf{A}} \\
&\leq 2\|\check{\mathbf{W}}_\omega^{-1}\mathbf{A}\|_{\mathbf{A}}\|\mathbf{A}^{-\frac{1}{2}}\mathbf{D}\mathbf{A}^{-\frac{1}{2}}\|_2 \\
&= 2\rho(\mathbf{D}\mathbf{A}^{-1})\|\mathbf{I}_n - \check{\mathbf{M}}_\omega\|_{\mathbf{A}} \leq 4\rho(\mathbf{D}\mathbf{A}^{-1})
\end{aligned}
\tag{38}
$$

we have by applying $\omega \in [1, \omega_{\max}]$ that

$$
\|\mathbf{A}^{-\frac{1}{2}}(\partial_\omega \check{\mathbf{C}}_\omega)\mathbf{A}^{\frac{1}{2}}\|_2 \leq \frac{2}{2-\omega} + \frac{2}{\omega} + \frac{8\rho(\mathbf{D}\mathbf{A}^{-1})}{\omega^2} \leq \frac{4}{2-\omega_{\max}} + 8\rho(\mathbf{D}\mathbf{A}^{-1})
\tag{39}
$$

$\square$

**Algorithm 8:** Symmetric successive over-relaxation with an absolute convergence condition.

**Input:** $\mathbf{A} \in \mathbb{R}^{n \times n}$, $\mathbf{b} \in \mathbb{R}^n$, parameter $\omega \in (0, 2)$, initial vector $\mathbf{x} \in \mathbb{R}^n$, tolerance $\varepsilon > 0$

$\mathbf{D} + \mathbf{L} + \mathbf{L}^T \leftarrow \mathbf{A}$     `// D` is diagonal, `L` is strictly lower triangular

$\mathbf{\breve{W}}_\omega \leftarrow \frac{\omega}{2-\omega}(\mathbf{D}/\omega + \mathbf{L})\mathbf{D}^{-1}(\mathbf{D}/\omega + \mathbf{L}^T)$     `// compute third normal form`

$\mathbf{r}_0 \leftarrow \mathbf{b} - \mathbf{A}\mathbf{x}$     `// compute initial residual`

**for** $k = 0, \dots$ **do**

    **if** $\|\mathbf{r}_k\|_2 > \varepsilon$ **then**

       **return** $k$     `// return iteration count (for use in learning)`

    $\mathbf{x} = \mathbf{x} + \mathbf{\breve{W}}_\omega^{-1}\mathbf{r}_k$ `// solve two triangular systems and update vector`

    $\mathbf{r}_{k+1} \leftarrow \mathbf{b} - \mathbf{A}\mathbf{x}$     `// compute the next residual`

**Output:** $k$

---

**Lemma F.2.** $\|\mathbf{\breve{C}}_\omega^k(c)\mathbf{b}\|_2$ *is* $\frac{10}{\lambda_{\min}(\mathbf{A})+c_{\min}}\rho(\mathbf{\breve{C}}_\omega)^{k-1}(c)\|\mathbf{b}\|_2 k\sqrt{\kappa(\mathbf{A})}$-*Lipschitz w.r.t. all* $c \geq c_{\min} > -\lambda_{\min}(\mathbf{A}(c))$, *where* $(c)$ *denotes matrices derived from* $\mathbf{A}(c) = \mathbf{A} + c\mathbf{I}_n$.

*Proof.* We take the derivative as in the above proof of Lemma F.1:

$$|\partial_c \|\mathbf{\breve{C}}_\omega^k(c)\mathbf{b}\|_2| = \rho(\mathbf{\breve{C}}_\omega(c))^{k-1}\|\mathbf{b}\|_2 k \|\mathbf{A}^{-\frac{1}{2}}(c)(\partial_c \mathbf{\breve{C}}_\omega(c))\mathbf{A}^{\frac{1}{2}}(c)\|_2 \sqrt{\kappa(\mathbf{A}(c))} \quad (40)$$

We then again apply the matrix calculus tool of Laue et al. (2018) to get

$$\begin{aligned}
\partial_c \mathbf{\breve{C}}_\omega(c) &= -\frac{2-\omega}{\omega}(\mathbf{D}(c)/\omega + \mathbf{L}^T)^{-1}\mathbf{D}(c)(\mathbf{D}(c)/\omega + \mathbf{L})^{-1} \\
&\quad + \frac{2-\omega}{\omega^2}\mathbf{A}(c)(\mathbf{D}(c)/\omega + \mathbf{L}^T)^{-2}\mathbf{D}(c)(\mathbf{D}(c)/\omega + \mathbf{L})^{-1} \\
&\quad - \frac{2-\omega}{\omega}\mathbf{A}(c)(\mathbf{D}(c)/\omega + \mathbf{L}^T)^{-1}(\mathbf{D}(c)/\omega + \mathbf{L})^{-1} \\
&\quad + \frac{2-\omega}{\omega^2}\mathbf{A}(c)(\mathbf{D}(c)/\omega + \mathbf{L}^T)^{-1}\mathbf{D}(c)(\mathbf{D}(c)/\omega + \mathbf{L})^{-2} \\
&= -\frac{2-\omega}{\omega}(\mathbf{\breve{W}}_\omega^{-1}(c) + \mathbf{A}(c)\mathbf{W}_\omega^{-T}(c)\mathbf{W}_\omega^{-1}(c)) \\
&\quad + \frac{2-\omega}{\omega^2}\mathbf{A}(c)(\mathbf{W}_\omega^{-T}(c)\mathbf{\breve{W}}_\omega^{-1}(c) + \mathbf{\breve{W}}_\omega^{-1}(c)\mathbf{W}_\omega^{-1}(c))
\end{aligned} \quad (41)$$

By symmetry of $\mathbf{\breve{W}}_\omega^{-1}(c)$ we have

$$\begin{aligned}
\|\mathbf{A}^{-\frac{1}{2}}(c)\mathbf{\breve{W}}_\omega^{-1}(c)\mathbf{A}^{\frac{1}{2}}(c)\|_2 = \|\mathbf{\breve{W}}_\omega^{-1}(c)\|_{\mathbf{A}(c)} &\leq \|\mathbf{\breve{W}}_\omega^{-1}(c)\mathbf{A}(c)\|_{\mathbf{A}(c)}\|\mathbf{A}^{-1}(c)\|_2 \\
&= \|\mathbf{I}_n - \mathbf{\breve{M}}_\omega(c)\|_{\mathbf{A}(c)}\rho(\mathbf{A}^{-1}(c)) \leq 2\rho(\mathbf{A}^{-1}(c))
\end{aligned} \quad (42)$$

Furthermore

$$\begin{aligned}
\|\mathbf{A}^{\frac{1}{2}}(c)\mathbf{W}_\omega^{-T}(c)\mathbf{W}_\omega^{-1}(c)\mathbf{A}^{\frac{1}{2}}(c)\|_2 &= \|\mathbf{A}^{-\frac{1}{2}}(c)(\mathbf{I}_n - \mathbf{M}_\omega^T(c))(\mathbf{I}_n - \mathbf{M}_\omega(c))\mathbf{A}^{-\frac{1}{2}}(c)\|_2 \\
&\leq \|\mathbf{A}^{-1}\|_2 \|\mathbf{I}_n - \mathbf{M}_\omega(c)\|_{\mathbf{A}(c)}^2 \leq 4\rho(\mathbf{A}^{-1}(c))
\end{aligned} \quad (43)$$

and

$$\|\mathbf{A}^{\frac{1}{2}}(c)\mathbf{W}_\omega^{-T}(c)\mathbf{\breve{W}}_\omega^{-1}(c)\mathbf{A}^{\frac{1}{2}}(c)\|_2 = \|\mathbf{A}^{\frac{1}{2}}(c)\mathbf{\breve{W}}_\omega^{-1}(c)\mathbf{W}_\omega^{-1}(c)\mathbf{A}^{\frac{1}{2}}(c)\|_2 \leq 4\rho(\mathbf{A}^{-1}(c)) \quad (44)$$

so by the lower bound of $\frac{1}{\lambda_{\min}(\mathbf{A})+c_{\min}}$ on $\rho(\mathbf{A}^{-1}(c))$ we have the result. $\square$

**Lemma F.3.** *Let $\mathbf{X} \in \mathbb{R}^{n \times n}$ be a nonzero matrix and $\mathbf{b} = m\mathbf{u}$ be a product of independent random variables $m \geq 0$ and $\mathbf{u} \in \mathbb{R}^n$ with $m^2 \in [0, n]$ a $\chi^2$-squared random variable with $n$ degrees of freedom truncated to the interval $[0, n]$ and $\mathbf{u}$ distributed uniformly on the surface of the unit sphere. Then for any interval $I = (\varepsilon, \varepsilon + \Delta] \subset \mathbb{R}$ for $\varepsilon, \Delta > 0$ we have that $\Pr(\|\mathbf{Xb}\|_2 \in I) \leq \frac{2\Delta}{\rho(\mathbf{X})}\sqrt{\frac{2}{\pi}}$.*

*Proof.* Let $f$ be the p.d.f. of $\mathbf{b}$ and $g$ be the p.d.f. of $\mathbf{g} \sim \mathcal{N}(\mathbf{0}_n, \mathbf{I}_n)$. Then by the law of total probability and the fact that $\mathbf{b}$ follows the distribution of $\mathbf{g}$ conditioned on $\|\mathbf{b}\|_2^2 \leq n$ we have that

$$
\begin{aligned}
\Pr(\|\mathbf{Xb}\|_2 \in I) &= \int_{\|\mathbf{x}\|_2^2 \leq n} \Pr(\|\mathbf{Xb}\|_2 \in I | \mathbf{b} = \mathbf{x}) df(\mathbf{x}) \\
&= \frac{\int_{\|\mathbf{x}\|_2^2 \leq n} \Pr(\|\mathbf{Xb}\|_2 \in I | \mathbf{b} = \mathbf{x}) dg(\mathbf{x})}{\int_{\|\mathbf{x}\|_2^2 > n} dg(\mathbf{x})} \\
&\leq 2 \int_{\|\mathbf{x}\|_2^2 \leq n} \Pr(\|\mathbf{Xg}\|_2 \in I | \mathbf{g} = \mathbf{x}) dg(\mathbf{x}) \\
&\leq 2 \int_{\mathbb{R}^n} \Pr(\|\mathbf{Xg}\|_2 \in I | \mathbf{g} = \mathbf{x}) dg(\mathbf{x}) = 2\Pr(\|\mathbf{Xg}\|_2 \in I)
\end{aligned}
\tag{45}
$$

where the second inequality uses the fact that a $\chi^2$ random variable with $n$ degrees of freedom has more than half of its mass below $n$. Defining the orthogonal diagonalization $\mathbf{Q}^T \Lambda \mathbf{Q} = \mathbf{X}^T \mathbf{X}$ and noting that $\mathbf{Qg} \sim \mathcal{N}(\mathbf{0}_n, \mathbf{I}_n)$, we then have that

$$
\|\mathbf{Xg}\|_2^2 = (\mathbf{Qg})^T \Lambda \mathbf{Qg} = \sum_{i=1}^n \Lambda_{[i,i]} \chi_i^2
\tag{46}
$$

for i.i.d. $\chi_1, \ldots, \chi_n \sim \mathcal{N}(0, 1)$. Let $h$, $h_1$, and $h_{-1}$ be the densities of $\sum_{i=1}^n \Lambda_{[i,i]} \chi_i^2$, $\Lambda_{[1,1]} \chi_1^2$, and $\sum_{i=2}^n \Lambda_{[i,i]} \chi_i^2$, respectively, and let $u(a)$ be the uniform measure on the interval $(a, a + 2\varepsilon\Delta + \Delta^2)$. Then since the density of the sum of independent random variables is their convolution, we can apply Young's inequality to obtain

$$
\begin{aligned}
\Pr(\|\mathbf{Xg}\|_2 \in I) = \Pr(\|\mathbf{Xg}\|_2^2 \in (\varepsilon^2, (\varepsilon + \Delta)^2]) \\
&\leq \max_{a \geq \varepsilon^2} \int_a^{a + 2\varepsilon\Delta + \Delta^2} h(x) dx \\
&= \max_{a \geq \varepsilon^2} \int_{-\infty}^{\infty} u(x - a) h(x) dx \\
&= \|u * h\|_{L^\infty([\varepsilon, \infty))} \\
&= \|u * h_1 * h_{-1}\|_{L^\infty([\varepsilon, \infty))} \\
&\leq \|u * h_1\|_{L^\infty([\varepsilon, \infty))} \|h_{-1}\|_{L^1([\varepsilon, \infty))} \\
&\leq \max_{a \geq \varepsilon^2} \int_a^{a + 2\varepsilon\Delta + \Delta^2} h_1(x) dx \\
&= \max_{a \geq \varepsilon} \int_a^{a + 2\varepsilon\Delta + \Delta^2} \frac{e^{-\frac{x}{2\Lambda_{[1,1]}}}}{\sqrt{2\pi \Lambda_{[i,i]} x}} dx \\
&\leq \max_{a \geq \varepsilon^2} \sqrt{\frac{2(a + 2\varepsilon\Delta + \Delta^2)}{\pi \Lambda_{[i,i]}}} - \sqrt{\frac{2a}{\pi \Lambda_{[i,i]}}} = \Delta\sqrt{\frac{2}{\pi \Lambda_{[i,i]}}}
\end{aligned}
\tag{47}
$$

Substituting into the first equation and using $\Lambda_{[i,i]} = \|\mathbf{X}\|_2^2 \geq \rho(\mathbf{X})^2$ yields the result. $\qquad\square$

**Lemma F.4.** *Suppose $\mathbf{b} = m\mathbf{u}$, where $m$ and $\mathbf{u}$ are independent random variables with $\mathbf{u}$ distributed uniformly on the surface of the unit sphere and $m^2 \in [0, n]$ a $\chi^2$-squared random variable with $n$ degrees of freedom truncated to the interval $[0, n]$. Define $K$ as in Corollary D.2, $\beta = \min_x \rho(\mathbf{I}_n - \mathbf{D}_x^{-1}\mathbf{A}_x)$, and $SSOR(x) = \min_{\|\check{\mathbf{C}}_x^k \mathbf{b}\|_2 \leq \varepsilon} k$ to be the number of iterations to convergence when the defect reduction matrix depends on some scalar $x \in \mathcal{X}$ for some bounded interval $\mathcal{X} \subset \mathbb{R}$. If $\|\check{\mathbf{C}}_x^k \mathbf{b}\|_2$ is $L\rho(\check{\mathbf{C}}_x)^{k-1}$-Lipschitz a.s. w.r.t. any $x \in \mathcal{X}$ then $\mathbb{E}SSOR$ is $\frac{32K^3 L\sqrt{2/\pi}}{\beta^4}$-Lipschitz w.r.t. $x$.*

*Proof.* First, note that by Hackbusch (2016, Theorem 6.26)

$$\rho(\check{\mathbf{C}}_x) = \rho(\check{\mathbf{M}}_x) = \|\check{\mathbf{M}}_x^2\|_{\mathbf{A}_x} \geq \rho(\mathbf{M}_x)^2 \geq \left(\frac{\beta}{1 + \sqrt{1-\beta^2}}\right)^4 \geq \frac{\beta^4}{16} \tag{48}$$

Now consider any $x_1, x_2 \in \mathcal{X}$ s.t. $|x_1 - x_2| \leq \frac{\varepsilon \beta^4 \sqrt{\pi/2}}{2K^3 L}$, assume w.l.o.g. that $x_1 < x_2$, and pick $x' \in [x_1, x_2]$ with maximal $\rho(\check{\mathbf{C}}_x)$. Then setting $\rho_{x'} = \rho(\check{\mathbf{C}}_{x'})$ we have that $\|\check{\mathbf{C}}_{x_i}^k \mathbf{b}\|_2$ is $L\rho_{x'}^{k-1}$-Lipschitz for both $i = 1, 2$ and all $k \in [K]$. Therefore starting with Jensen's inequality we have that

$$
\begin{aligned}
&|\mathbb{E}SSOR(x_i) - \mathbb{E}SSOR(x')| \\
&\leq \mathbb{E}|SSOR(x_i) - SSOR(x')| \\
&= \sum_{k=1}^{K} \sum_{l=1}^{K} |k - l| \Pr(SSOR(x_i) = k \cap SSOR(x') = l) \\
&\leq K \sum_{k=1}^{K} \left( \sum_{l<k} \Pr(\|\check{\mathbf{C}}_{x_i}^l \mathbf{b}\|_2 > \varepsilon \cap \|\check{\mathbf{C}}_{x'}^l \mathbf{b}\|_2 \leq \varepsilon) + \sum_{l>k} \Pr(\|\check{\mathbf{C}}_{x_i}^k \mathbf{b}\|_2 \leq \varepsilon \cap \|\check{\mathbf{C}}_{x'}^k \mathbf{b}\|_2 > \varepsilon) \right) \\
&\leq K \sum_{k=1}^{K} \sum_{l<k} \Pr(\|\check{\mathbf{C}}_{x'}^l \mathbf{b}\|_2 \in (\varepsilon - L\rho_{x'}^{l-1}|x_i - x'|, \varepsilon]) \\
&\quad + K \sum_{k=1}^{K} \sum_{l>k} \Pr(\|\check{\mathbf{C}}_{x'}^k \mathbf{b}\|_2 \in (\varepsilon, \varepsilon + L\rho_{x'}^{k-1}|x_i - x'|]) \\
&\leq K \sum_{k=1}^{K} \left( \sum_{l<k} \frac{2L\rho_{x'}^{l-1}\sqrt{2/\pi}}{\rho(\check{\mathbf{C}}_{x'}^l)}|x_i - x'| + \sum_{k=1}^{K} \sum_{l>k} \frac{2L\rho_{x'}^{k-1}\sqrt{2/\pi}}{\rho(\check{\mathbf{C}}_{x'}^k)}|x_i - x'| \right) \\
&\leq \frac{2K^3 L\sqrt{2/\pi}}{\rho_{x'}}|x_i - x'| \leq \frac{32K^3 L\sqrt{2/\pi}}{\beta^4}
\end{aligned}
$$
$$\tag{49}$$

where the second inequality follows by the definition of $SSOR$, the third by Lipschitzness, and the fourth by the anti-concentration result of Lemma F.3. Since this holds for any nearby pairs $x_1 < x_2$, taking the summation over the interval $\mathcal{X}$ completes the proof. $\qquad\square$

**Corollary F.1.** *Under the assumptions of Lemma E.1, the function $\mathbb{E}_\mathbf{b} SSOR(\mathbf{A}, \mathbf{b}, \omega)$ is $\frac{32K^4\sqrt{2n\kappa(\mathbf{A})/\pi}}{\beta^4}\left(\frac{1}{2-\omega_{\max}} + 2\rho(\mathbf{DA}^{-1})\right)$-Lipschitz w.r.t. $\omega \in [1, \omega_{\max}] \subset (0, 2)$.*

*Proof.* Apply Lemmas F.1 and E.1, noting that $\|\mathbf{b}\|_2 \leq \sqrt{n}$ by definition. $\qquad\square$

**Corollary F.2.** *Under the assumptions of Lemma E.1, the function $\mathbb{E}_\mathbf{b} SSOR(\mathbf{A}(c), \mathbf{b}, \omega)$ is $\max_c \frac{320K^4\sqrt{2n\kappa(\mathbf{A}(c))/\pi}}{\beta^4(\lambda_{\min}(\mathbf{A})+c_{\min})}$-Lipschitz w.r.t. $c \geq c_{\min} > -\lambda_{\min}$.*

*Proof.* Apply Lemma F.2 and E.1, noting that $\|\mathbf{b}\|_2 \leq \sqrt{n}$ by definition. $\qquad\square$

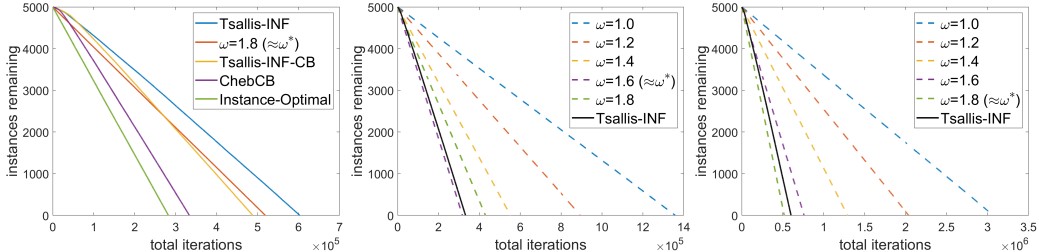

Figure 5: Average across forty trials of the time needed to solve 5K diagonally shifted systems with $\mathbf{A}_t = \mathbf{A} + \frac{12c-3}{20}\mathbf{I}_n$ for $c \sim \text{Beta}(\frac{1}{2}, \frac{3}{2})$ (center) and $c \sim \text{Beta}(2, 6)$ (otherwise).

## G EXPERIMENTAL DETAILS

All numerical results were generated in MATLAB on a laptop and can be re-generated by running the scripts available at https://github.com/mkhodak/learning-to-relax. Note that, since we do not have access to problem parameters, we experimented with a few approaches to setting them automatically or heuristically on the simplest (low variance) setting below and then used the same settings for the rest of the experiments (high variance and heat equation). Furthermore, because the default step-size/learning rate settings in both algorithms are rather pessimistic, we use more aggressive time-varying approaches in practice. For Tsallis-INF we set $\eta_t = 2/\sqrt{t}$, which is what is used in the anytime variant (Zimmert & Seldin, 2021). As for ChebCB, we use an increasing schedule $\eta_t = \mathcal{O}(t)$; note that Simchi-Levi & Xu (2021) also use an increasing learning rate schedule for setting inverse gap-weighted probabilities.

### G.1 BASIC EXPERIMENTS

For the experiments in Figure 2 (center-left), we sample $T = 5K$ scalars $c_t \sim \text{Beta}(2, 6)$ and run Tsallis-INF, Tsallis-INF-CB ChebCB, the instance optimal policy $\omega^*(c)$, and five values of $\omega$—evenly spaced on $[1, 1.8]$—on all instances $\mathbf{A}_t = \mathbf{A} + \frac{12c_t-3}{20}\mathbf{I}_n$, in random order. Note that Figure 5 (left) contains results of the same setup, except with $c_t \sim \text{Beta}(\frac{1}{2}, \frac{3}{2})$, the higher-variance setting from Figure 1. The center and right figures contain results for the sub-optimal fixed $\omega$ parameters, compared to Tsallis-INF. For both experiments the matrix $\mathbf{A}$ is again the $100 \times 100$ Laplacian of a square-shaped domain generated in MATLAB, and the targets $\mathbf{b}$ are re-sampled at each instance from the Gaussian truncated radially to have norm $\leq n$. The reported results are averaged of forty trials.

### G.2 ACCELERATING A 2D HEAT EQUATION SOLVER

We then consider applying our methods to the task of numerical simulation of the 2D heat equation

$$\partial_t u(t, \mathbf{x}) = \kappa(t)\Delta_{\mathbf{x}} u(t, \mathbf{x}) + f(t, \mathbf{x}) \tag{50}$$

over the domain $\mathbf{x} \in [0, 1]^2$ and $t \in [0, 5]$. We use a five-point finite difference discretization with size denoted $n_{\mathbf{x}} = 1/\Delta_{\mathbf{x}}$, so that when an implicit time-stepping method such as Crank-Nicolson is applied with timestep $\Delta_t$ the numerical simulation requires sequentially solving a sequence of linear systems $(\mathbf{A}_t, \mathbf{b}_t)$ with $\mathbf{A}_t = \mathbf{I}_{(n_{\mathbf{x}}-1)^2} - \kappa((t + 1/2)\Delta_t)\mathbf{A}$ for a fixed matrix $\mathbf{A}$ (that depends on $\Delta_t$ and $\Delta_{\mathbf{x}}$) corresponding to the discrete Laplacian of the system (LeVeque, 2007, Equation 12.29). Each $\mathbf{A}_t$ is positive definite, and moreover note that mathematically the setting is equivalent to an instantiation of the diagonal offset setting introduced in Section 2.4, since the linear system is equivalent to $c_t\mathbf{I}_{(n_{\mathbf{x}}-1)^2} - \mathbf{A} = c_t\mathbf{b}_t$ for $c_t = 1/\kappa((t + 1/2)\Delta_t)$. However, for simplicity we will simply pass $\kappa((t + 1/2\Delta_t))$ as contexts to CB methods.

To complete the problem specification, define the *bump function* $b_{\mathbf{c},r}(\mathbf{x})$ centered at $\mathbf{c} \in \mathbb{R}^2$ with radius $r > 0$ to be $\exp\left(-\frac{1}{1-\|\mathbf{x}-\mathbf{c}\|_2^2/r^2}\right)$ if $\|\mathbf{x}-\mathbf{c}\|_2 < r$ and $0$ otherwise. We set the initial condition $u(0, \mathbf{x}) = b_{(\frac{1}{2} \frac{1}{2}), \frac{1}{4}}(\mathbf{x})$, forcing function $f(t, \mathbf{x}) = 32b_{(\frac{1}{2}+\cos(16\pi t)/4, \frac{1}{2}+\cos(16\pi t)/4), 1/8}(\mathbf{x})$, and diffusion coefficient $\kappa(t) = \max\{0.01\sin(2\pi t), -10\sin(2\pi t)\}$. The forcing function—effectively a bump circling around the center of the domain—is chosen to ensure that the linear system solutions

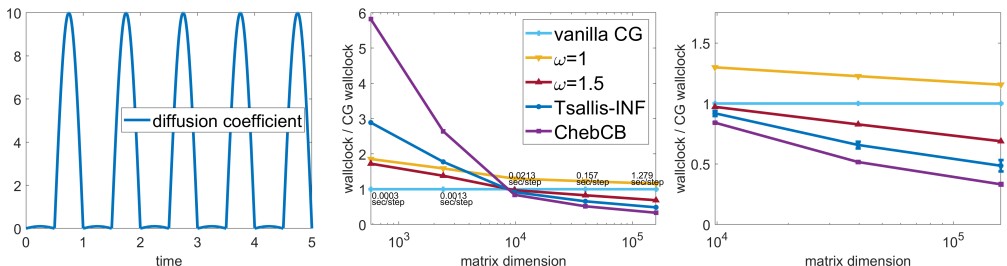

Figure 6: Diffusion coefficient as a function of time (left) and normalized total wallclock time required to run 5K steps of the numerical simulation (center and right). The numbers within the middle plot corresponding to the average number of seconds required to run a step of the simulation using vanilla CG. The right-hand plot shows 95% confidence intervals across the three trials for Tsallis-INF and ChebCB at the three higher-dimensional evaluations.

are not too close to each other or to zero, and the diffusion coefficient function—plotted in Figure 6 (left)—is chosen to make the instance-optimal $\omega$ behave roughly periodically (c.f. Figure 2 (right)).

We set $\Delta_t = 10^{-3}$, thus making $T = 5000$, and evaluate our approach across five spatial discretizations: $n_{\mathbf{x}} = 25, 50, 100, 200, 400$. The resulting linear systems have size $n = (n_{\mathbf{x}} - 1)^2$. At each timestep, we solve each linear system using CG to relative precision $\varepsilon = 10^{-8}$. The baselines we consider are vanilla (unpreconditioned) CG and SSOR-CG with $\omega = 1$ or $\omega = 1.5$; as comparators we also evaluate performance when using the best fixed $\omega$ in hindsight at each round, and when using the *instance-optimal* $\omega$ at each round. Recall that we showed that Tsallis-INF has sublinear regret w.r.t. the surrogate cost of the best fixed $\omega$ of SSOR-CG (Theorem 2.3), and that ChebCB has sublinear regret w.r.t. the instance-optimal $\omega$ for SOR in the semi-stochastic setting of Section 3. Since both methods are randomized, we take the average of three runs.

In Figure 2 (center-right) we show that both methods substantially outperform all three baselines, except at $n_{\mathbf{x}} = 25$ and $n_{\mathbf{x}} = 50$, when $\omega = 1.5$ almost recovers the best fixed parameter in hindsight; furthermore, ChebCB does better then the best fixed $\omega$ in hindsight in most cases. In Figure 6 (right) we also show that—at high-enough dimensions—this reduction in the number of iterations leads to an overall improvement in the *runtime* of the simulation. Several other pertinent notes include:

1. At lower dimensions the learning-based approaches have slower overall runtime because of overhead associated with learning; ChebCB in particular solves a small constrained linear regression at each step. However, this overhead does *not* scale with matrix dimension, and we expect data-driven approaches to have the greatest impact in higher dimensions.

2. Vanilla (unpreconditioned) CG is faster than SSOR-preconditioned CG with $\omega = 1$ despite having more iterations because each iteration is more costly.

3. To get a comparative sense of the scale of the improvement, we can consider the results in Li et al. (2023, Table 1), who learn a (deep-learning-based) preconditioner for CG to simulate the 2D heat equation. In the precision $10^{-8}$ case their solver takes 2.3 seconds, while Gauss-Seidel (i.e. SSOR with $\omega = 1$) takes 2.995 seconds, a roughly 1.3x improvement. In our most closely comparable setting, Tsallis-INF and ChebCB are roughly 2.4x and 3.5x faster than Gauss-Seidel, respectively (and have other advantages such as simplicity and being deployable in an online fashion without pretraining). We caveat this comparison by noting that Li et al. (2023) consider a statistical, not online, learning setup, and their matrix structure may be significantly different—it results from a finite element method rather than finite differences. The only way to achieve a direct comparisons is via access to code; as of this writing it is not public.

Lastly, we give additional details for the plot in Figure 2 (right), which shows the actions taken by the various algorithms for a simulation at $n_{\mathbf{x}} = 100$. For clarity all lines are smoothed using a moving average with a window of 25, and for Tsallis-INF and ChebCB we also shade $\pm$ one standard deviation computed over this window. The plot shows that Tsallis-INF converges to an action close to the best fixed $\omega$ in hindsight, and that ChebCB fairly quickly follows the instance-optimal path, with the standard deviation of both decreasing over time.

