# OpenReview forum: "Learning to Relax: Setting Solver Parameters Across a Sequence of Linear System Instances"
_ICLR.cc/2024/Conference — ICLR 2024 spotlight_

### Official Review · Reviewer_655G · 2023-10-30

**Soundness:** 3 good
**Presentation:** 3 good
**Contribution:** 3 good
**Rating:** 8
**Confidence:** 2

**Summary:**

This paper establishes learnability result for the overrelaxation parameter in SOR method. Online learning algorithm and regret bounds are analyzed for solving a sequence of linear systems.

**Strengths:**

The paper theoretically justifies that algorithm parameter in linear system solving is learnable. Online algorithm with provable regret guarantees is given. The context setup and analysis are novel to my knowledge.

**Weaknesses:**

The paper adopts an upperbound surrogate loss function. Although this is reasonable due to the hardness in characterizing instance-specific convergence behavior of iterative methods, this still results in potential gap between theory and practice.

**Questions:**

1. This paper focuses on online learning algorithms. In practice this may still be inefficient in the exploration phase. Is it possible to adopt a static learning model and prove learnability result using tools from statistical learning theory? (e.g., train a classifier/regressor to predict the best $\omega$)
2. The paper assumes iterative methods start from scratch. When solving a sequence of linear systems that are potentially from the same distribution, a common practice is to warm-start with previous solutions, and it often speeds up convergence. Is it possible also to incorporate this into your analysis?

**Minor typos and stylistic issues**

1. Page 14

   guarantes => guarantees

2. Page 26

   folloows => follows

---

> ### Author Response · Authors · 2023-11-20
> **Response to Reviewer 655G**
>
> Thank you for your positive review! We hope to address your questions below:
>
> 1. [*The paper adopts an upper-bound surrogate loss function. Although this is reasonable due to the hardness in characterizing instance-specific convergence behavior of iterative methods, this still results in potential gap between theory and practice.*]
> This is indeed a limitation of the results, although note we also study the non-surrogate case in Sec. 3 under a “semi-stochastic” assumption; please also see our section “On assumptions” in the [general response](https://openreview.net/forum?id=5t57omGVMw&noteId=rqYP3IrU0O) for a more thorough discussion.
> 2. [*This paper focuses on online learning algorithms. In practice this may still be inefficient in the exploration phase. Is it possible to adopt a static learning model and prove learnability result using tools from statistical learning theory? (e.g., train a classifier/regressor to predict the best.*]
> Yes, while our paper focuses on online learning in the main part we do show sample complexity bounds obtained by doing offline training in App. A.1. Our approach leads to comparable bounds to those that can be obtained by directly applying the GJ framework (Bartlett et al., 2022), but unlike them we also provide an efficient learning algorithm to do it.
> 3. [*The paper assumes iterative methods start from scratch. When solving a sequence of linear systems that are potentially from the same distribution, a common practice is to warm-start with previous solutions, and it often speeds up convergence. Is it possible also to incorporate this into your analysis?*]
> Yes, the near-asymptotic results can handle a warm-started vector; in fact if ${\bf x}_t$ is the warm-start vector at step $t$ it is as simple as replacing ${\bf b}_t$ by ${\bf b}_t-{\bf A}_t{\bf x}_t$ when defining $\tau_t$ via Ass. 2.1. Note that this does assume the warm-starts do not depend on previous settings of $\omega$, which is not strictly true; however, the dependence is quite small ($O(\varepsilon)$).
>
> # References
> Bartlett, Indyk, Wagner. *Generalization bounds for data-driven numerical linear algebra*. COLT 2022.

---

> > ### Comment · Reviewer_655G · 2023-11-20
> >
> > Thank you for the response, and I keep my positive evaluation of the paper.

---

### Official Review · Reviewer_vVy3 · 2023-10-31

**Soundness:** 3 good
**Presentation:** 3 good
**Contribution:** 4 excellent
**Rating:** 8
**Confidence:** 3

**Summary:**

In this paper, the authors study different settings of sequentially solving a series of (related) linear systems using Successive-Over-Relaxation (SOR). The problem is that the runtime/number of iterations of SOR highly depends on the "relaxation parameter" $\omega$, so the challenge is to learn online good choices of $\omega$. They look at two settings. First, a deterministic setting where we assume the instances are close enough to the "asymptotic regime" where the convergence rate of SOR decreases linearly with respect to the spectral radius of a specific problem-dependent matrix. The second setting we assume the target vector is randomly sampled from what we can think as a truncated Gaussian distribution. In both cases, they show how bandit algorithms can be used to pick parameters $\omega$ online that work well on average. Furthermore, they show how to use contextual bandits in the case when we have more structured information about the linear systems we need to solve.

**Strengths:**

- This is a paper with a very solid contribution on data-driven algorithm design (and a few interesting technical bits for adversarial badits), with interesting new ideas applied to a simple but interesting problem, and I believe will likely lead to interesting follow-up work.

    More specifically, generalizing bandits algorithms in the adversarial setting for functions that are not exactly Lipschitz continuous for the use in algorithms with predictions/data-driven algorithm design is very interesting and seems to work quite well. The extension to the use of Contextual Bandit algorithms when we are looking at a series of systems with shifted diagonals is interesting, and how we can exploit the stochastic case and skip the use of a surrogate function at all are quite interesting.

    Even if some of the bounds do not seem to lead to very informative regret guarantees, it seems to be full of interesting technical ideas and does a good job of showing what are the difficulties of controlling the dependency of problem-parameters in these regret bounds.

- As the authors themselves mention, this is an interesting case where they designed algorithms of two related lines of work (algorithms with predictions and data-driven algorithm design)

- The authors are very careful to not over-sell their contributions: caveats and weaknesses are often immediately mentioned and discussed, together with a summary of these drawbacks at the conclusions sections;
- The presentation is not perfect, but very good considering how much ground the authors cover in 9 pages. I believe this paper covered a lot of ground, and after skimming a few parts of the appendix it does seem to be very thorough
- Although it is in the appendix due to space limitations, I really enjoyed reading the related work section since I lack the background in linear system solvers and the related literature;

**Weaknesses:**

- As mentioned by the authors, some of performance guarantees seem to be very loose (mainly the ones in section 3 with the dependency on dimension);
- I am more acquainted with the online learning and optimization literature, and probably did not have enough background to judge how good some of the bounds were (mainly, the dependency on problem parameters in Thm 2.2 and 2.3). Although this is a problem with my lack of background, I believe the authors could add a bit of discussion of what would be "ideal" (?) dependencies here;
- It seems to be that having stochastic target vectors with independent entries is a very strong assumption. The authors acknowledge that assuming the vectors come from a (scaled and somewhat truncated) Gaussian is restrictive, but it was not clear if mild correlation between the entries (positive definite correlation matrix with small off-diagonal entries?) would make the analysis break down;
- Although the experiments are not the focus of the paper, I do believe they are a bit too simplistic. I do no think the authors should spend time coming up with more experiments, but I think that discussing the limitations of the experiments would be helpful.


I am not an expert in algorithms with predictions/data-drive algorithm design (I'm more acquainted with online learning and optimization), so I will take into account the other reviews for my final scores. I might be willing to increase my score, the only reasons why I haven't said yet that this should be highlighted is because ICLR is a broad conference (with a slight focus on deep learning and other large-scale ML techniques), so I am not sure if I would say this should be highlighted, but the contributions are definitely substantive and solid.

**Questions:**

- A few times the authors mention that they provide "end-to-end" guarantees, but I am not sure if I can parse what you mean by this. Could your briefly mention that do you mean by this?
- If the authors have time, could you expand a bit on the poor dependency on the spectral properties of the instances in theorem 2.3 (and maybe 2.2)? The authors mention that these dependencies are not ideal, but I do not have enough context to know what are dependencies that would be more "acceptable";
- On the assumption on $b_t$'s in sec 3, is it really necessary for the entries to be independent of each other? Moreover, how much do you depend on $b_t$'s being exactly the distribution you have? I probably could answer this questions myself if I had the time to go carefully over the proofs, but if the authors could briefly comment on this, it would be great;

### Suggestions
- The discussion on "asymptocity" in sec 2.2 is very unclear. At some point I think I understood more or less what you meant, but this is a very confusing part early on in the paper, and there is not enough context to interpret the plot. Maybe this is more of a note, but if you could expand on this in a revised version of the paper, it would be great.
- In Theorem 2.1 you cite "Lemma 2.1.2" to mean item 2 of Lemma 2.1, which is a bit weird since this is easily misunderstood as a separate lemma (that does not exist);
- Although I mentioned about one of the plots in Fig 2 before, I think all of the plots in Fig 2 need more context. It is not clear to me what the middle plot is trying to show, and the last plot is great but I only managed to understand that all of the algorithms there were algorithms proposed in the paper by the end of the paper. So when I first read this plot I was very confused;

---

> ### Author Response · Authors · 2023-11-20
> **Response to Reviewer vVy3**
>
> Thank you for your positive review! We appreciate the thoughtful suggestions and hope to address some of your concerns below:
>
> 1. [*As mentioned by the authors, some of performance guarantees seem to be very loose (mainly the ones in section 3 with the dependency on dimension)*]
> Yes, especially for the Sec. 3 results we might hope for logarithmic or better dependence on the dimension. At the same time, we were quite happy to get a logarithmic dependence on $1/\varepsilon$, which we view as important and non-trivial.
> 2. [*could you expand a bit on the poor dependency on the spectral properties of the instances in theorem 2.3 (and maybe 2.2)? The authors mention that these dependencies are not ideal, but I do not have enough context to know what are dependencies that would be more "acceptable”*]
> Apologies: we assume by “not ideal” you are referring to what we say in the first sentence of the last paragraph of the Conclusion. The phrasing there is a holdover from when Sec. 2 only had one main result. In fact we view all of the results in Sec. 2 as having “reasonable” dependence on spectral quantities (although it is unclear what the “correct” dependence is) while the results in Sec. 3 are perhaps less reasonable because of high (4th-order) powers in various spectral quantities. At the same time, we do view the main theoretical challenge as overcoming the dimension-dependence in Sec. 3.
> 3. [*It seems to be that having stochastic target vectors with independent entries is a very strong assumption [...] it was not clear if mild correlation between the entries (positive definite correlation matrix with small off-diagonal entries?) would make the analysis break down [...] Moreover, how much do you depend on $b_t$’s being exactly the distribution you have?*]
> In general we agree that it is restrictive, but at the same time we do view the stochastic target vector assumption as a fairly reasonable simplification; please see the second paragraph in the “On assumptions” section in our [general response](https://openreview.net/forum?id=5t57omGVMw&noteId=rqYP3IrU0O) for a more thorough discussion. On whether one could extend the analysis to general (truncated) Gaussians: we believe that this might be doable, and that the resulting Lipschitz constant (and thus also the regret bound) might be the existing one multiplied by $O(1/\lambda_{\min}(\Sigma))$, i.e. it will be worse by roughly a factor of the reciprocal of the smallest eigenvalue of the covariance matrix. On whether there is an extension to more general distributions: Gaussianity is crucial to the anti-concentration component of the Lipschitz result ((46) and (47)), but perhaps the conversion from the truncated Gaussian in (45) can be generalized to conversions from other distributions.
> 4. [*Although the experiments are not the focus of the paper, I do believe they are a bit too simplistic. I do no think the authors should spend time coming up with more experiments, but I think that discussing the limitations of the experiments would be helpful.*]
> We had already been thinking about additional experiments and so have included them in the update; please see the [general response](https://openreview.net/forum?id=5t57omGVMw&noteId=rqYP3IrU0O) for more details. While limitations naturally remain—the variant of the heat equation being solved does not obviously correspond to a specific known task, and the simulation is not truly large-scale—the sequence of linear systems is generated by an actual numerical simulation, and the size of the matrices is much larger (are high as $n\ge100,000$).
> 5. [*A few times the authors mention that they provide "end-to-end" guarantees [...] Could your briefly mention that do you mean by this?*]
> By “end-to-end guarantees” we mean guarantees that consider the full learning pipeline, from collecting data to optimizing to running the learned algorithm. In particular, end-to-end guarantees do *not* just treat learned advice as an oracle without considering whether it can be learned from data (as is often in the case in the algorithms with predictions literature) and do *not* just show learnability of an algorithm without providing a computationally efficient learning algorithm (as is often the case in the data-driven algorithm design literature). Note that we do provide a brief definition of this on the second line of page 2, although we can expand upon that.
> 6. [*Suggestions*]
> Thank you for the clarity suggestions: we will continue working on the descriptions of “asymptocity” and of the figures, and we have fixed the Lemma 2.1 reference. In Fig. 2 (left in the update, center before), we aim to show how by using a degenerate target, we can make the objective be optimal at and vary strongly around a specific $\omega$ (here we do $\omega=1.4$ by setting ${\bf b}$ to the be the smallest eigenvector of the iteration matrix corresponding to $\omega=1.4$). This suggests that assumptions on ${\bf b}$ might be necessary for learning.

---

> > ### Comment · Reviewer_vVy3 · 2023-11-22
> > **Great response**
> >
> > I'd like to thank the authors for such a great response (both to my review and the general response). I'm really sorry that I did not engage too much in discussion during the discussion period. I read your response as soon as you posted it, but since I did not have any pressing questions or remarks, I decided to focus on other borderline papers in my batch.
> >
> > Overall the authors did an amazing job answering my questions, even if a couple of my questions or somewhat confusing (I'm sorry about it). The replies really made me appreciate the paper even more. During the reviewer discussion period I'll likely strongly vouch for the acceptance of this paper.

---

> > > ### Author Response · Authors · 2023-11-23
> > > **Thank you**
> > >
> > > Thank you for the kind words and your reviewing efforts!

---

### Official Review · Reviewer_7bs2 · 2023-11-01

**Soundness:** 3 good
**Presentation:** 4 excellent
**Contribution:** 3 good
**Rating:** 8
**Confidence:** 3

**Summary:**

This a paper about learning a good choice of a parameter of a linear system solver in an online (bandit-like) fashion. The goal is to minimize the number of iterations (and, as a consequence, the running time) of the solver. The paper is theoretical in nature – the authors prove regret bounds; simple proof-of-principle-style experiments serve only a complementary role.

The solver of choice for the paper is Successive Over-Relaxation (SOR). It is not the most widely used in practice solver nowadays, but it has some uses, and it has the advantage of being simple and easy to analyse. The authors mention extending their theory to more complex solver as an important direction of future research.

SOR is an iterative solver, and it has a single parameter, denoted by omega. The authors study the following setting: a sequence of linear systems is solved one by one. Before we start solving each system we can pick a value of omega, and after the system is solved we only learn for how many iterations the solver was running.

The authors propose to use Tsallis-INF bandit algorithm (a variant of more popular Exp3) to learn the parameter omega. The challenge lies in analysis – the number of iterations as a function of the parameter is non-Lipschitz. Instead, the authors use a continuous surrogate upper bound on the number of iterations, which they claim is reasonably tight. Unfortunately, this means they can only prove that the surrogate cost of their algorithm is close (up to a sublinear regret) to optimal surrogate cost (not the real cost, which might be much smaller) for a fixed parameter chosen in hindsight.

The second setting studied is in the spirit of contextual bandits. The authors assume all coefficient matrices are the same up to a linear shift of the diagonal. They show that under this assumption it is possible to obtain sublinear regret with respect to best choice of the parameter for each value of the shift separately. Since for a fixed shift there is a fixed optimal parameter, it means that the benchmark is just the optimal unconstrained choice of parameter. This is a very strong result (though only under a strong assumption about input, and still about the surrogate cost).

Finally, the authors also study a setting where target vectors (but not coefficient matrices) are drawn from a distribution, and in this setting they are able to show sublinear regret bounds using the actual (not the surrogate) cost of the algorithm.

**Strengths:**

The setting studied is very natural and this paper can easily stimulate further research in the area.

The algorithm used is a standard one, so the proposed approach seems more practical than if it was an ad-hoc algorithm designed specifically so that the analysis works.

The paper is nicely written – the authors explain why they do certain things instead of just presenting proofs out of the blue.

It seems that the technical content is novel and nontrivial – though I do not know the area well enough to be certain about that.

**Weaknesses:**

The results are either about the surrogate loss, or under the assumption that part of the input is stochastic and not adversarial.

Experiments are very basic. It is not a huge weakness as this paper can be positioned as a pure theory paper. However, since the solver seems to be used in practice, it seems it would not be difficult to get some actual input data it is run on, and results of such experiments would be very interesting to see.

**Questions:**

Do you have any (even soft) argument that the limitations imposed (surrogate loss or stochastic input) are necessary to prove sublinear regret bounds?

It seems that learning optimal parameters for iterative methods (LP solvers, linear systems solvers, GD, etc.) should be similar and explainable with a single theory. Do you have any understanding why it does not seem to be the case, and we have separate results for each of these methods?

How your work compares to what people do in practice? Is it clear why they are not using similar methods? Would it be feasible to compare empirical performance of your approach to what is already done in practice?

Minor remarks:

Abstract: "we prove that a bandit algorithm (...) can select parameters (...) such that the overall cost is almost as good as that the best fixed \omega would have obtained" – this sentence is technically not true, you prove it only for the surrogate upper bound and not for the actual cost

Page 2: "ir" -> "it", "in-addition" -> "in addition"

Page 4: "known scalars c_t" – I'm not sure what "known" means here; even if the scalars are not given explicitly, they can be trivially inferred from the input.

Page 4: please remind the reader that rho denotes the spectral radius – it might be clear for optimization people but not necessarily for learning people.

Page 14: ",e.g." -> ", e.g."

---

> ### Author Response · Authors · 2023-11-20
> **Response to Reviewer 7bs2**
>
> Thank you for your positive review! We agree that the setting lends itself to future work, and hope to address some of your concerns below:
>
> 1. [*The results are either about the surrogate loss, or under the assumption that part of the input is stochastic and not adversarial.*]
> This is indeed a limitation of the work; please also see our section “On assumptions” in the [general response](https://openreview.net/forum?id=5t57omGVMw&noteId=rqYP3IrU0O) for a more thorough discussion.
> 2. [*Experiments are very basic. It is not a huge weakness as this paper can be positioned as a pure theory paper. However, since the solver seems to be used in practice, it seems it would not be difficult to get some actual input data it is run on, and results of such experiments would be very interesting to see.*]
> We have updated our paper to include experiments where the sequence of linear systems is generated by an actual numerical simulation, and the size of the matrices is much larger (as high as $n\ge100,000$). Please see our [general response](https://openreview.net/forum?id=5t57omGVMw&noteId=rqYP3IrU0O) for a summary and discussion of these results.
> 3. [*Do you have any (even soft) argument that the limitations imposed (surrogate loss or stochastic input) are necessary to prove sublinear regret bounds?*]
> Please see the second half of the second paragraph in the “On assumptions” section of the general response for a soft argument on the possible necessity of some type of restriction. There we note that Fig. 2 (left) shows that degenerate target vectors can generate objective functions reminiscent of those used to show lower bounds in online learning.
> 4. [*It seems that learning optimal parameters for iterative methods (LP solvers, linear systems solvers, GD, etc.) should be similar and explainable with a single theory. Do you have any understanding why it does not seem to be the case, and we have separate results for each of these methods?*]
> Interesting question. If one is mainly interested in sample complexity bounds then there do exist very general frameworks (e.g. dual function analysis (Balcan et al., 2021) or GJ (Bartlett et al., 2022)) that likely provide guarantees for some of these algorithms (we make a direct comparison to GJ in App. A.1). However, they do not always yield efficient algorithms or regret guarantees, which is what we were mainly interested in. While we are not familiar enough with LP solvers to know the relevant tunable parameters (barrier coefficient schedules for interior point methods?) and their effect, the difficulty of adjusting a continuous parameter for a discontinuous response is that one then has to make sure the discontinuities are controlled somehow (otherwise learning is hard since learning thresholds is hard). As a result, proofs end up depending on properties of the algorithm that make them behave reasonably but do not extend to other algorithms; for example, the step-size tuning result of Gupta & Roughgarden (2017) depends crucially on the guaranteed progress of GD, which does not hold for SOR. Similarly, our results in Sec. 3 depend on each iterate being a deterministic matrix power times a random vector; however, in Chebyshev semi-iteration each iterate is a matrix polynomial multiplied by a vector, which is also the case for conjugate gradient except now the polynomial coefficients depend on the vector as well. Thus studying iterative algorithms has so far involved making technical leaps, but perhaps a more general framework exists here as well.
> 5. [*How your work compares to what people do in practice? Is it clear why they are not using similar methods? Would it be feasible to compare empirical performance of your approach to what is already done in practice?*]
> When SOR and SSOR-preconditioned CG are used in practice, $\omega$ is usually set to a fixed parameter such as 1 or 1.5. Methods similar to those proposed may not yet be being used because of a preference for simplicity or a concern that learning is not robust. In our updated paper, we include experiments that compare to a setting more similar to what is done in practice (c.f. the [general response](https://openreview.net/forum?id=5t57omGVMw&noteId=rqYP3IrU0O)).
> 6. [*Minor remarks*]
> Thank you for these notes; we have fixed all of them except the one about the abstract, as while the statement is not technically true in the near-asymptotic setting (Sec. 2), it *is* true in the semi-stochastic setting (Sec. 3).
>
> # References
> Balcan, DeBlasio, Dick, Kingsford, Sandholm, Vitercik. *How much data is sufficient to learn high-performing algorithms? Generalization guarantees for data-driven algorithm design*. STOC 2021.
> Bartlett, Indyk, Wagner. *Generalization bounds for data-driven numerical linear algebra*. COLT 2022.
> Gupta, Roughgarden. *A PAC approach to application-specific algorithm selection*. SIAM J. Computing, 2017.

---

### Official Review · Reviewer_9VfQ · 2023-11-04

**Soundness:** 3 good
**Presentation:** 3 good
**Contribution:** 3 good
**Rating:** 8
**Confidence:** 3

**Summary:**

The paper considers an online parameter optimization problem for sequentially solving linear system equations with a specific iterative algorithm (SOR) with a parameter.  The goal is formulated as a regret minimization where each cost per trial is defined as the number of iterations with the parameter at the trial. The critical observation is that the cost is further upper bounded by a convex surrogate function and analyzed under the online convex optimization framework.

**Strengths:**

The problem is well-motivated (even though the algorithm looks slightly restrictive). The critical observation is that the SOR iteration is bounded by a convex surrogate function with the parameter. The observation is non-trivial and thus the paper shows a new and interesting application of the online convex optimization framework.

**Weaknesses:**

Maybe a weakness of the paper is that the reduction to OCO is restricted to a certain type of algorithm (SOR) only so far. But I do not think the weakness is not so crucial since it shows, to the best of my knowledge, a new application of numerical optimization from OCO.

**Questions:**

Is it possible to extend this framework to other algorithms for solving linear equations?

---

> ### Author Response · Authors · 2023-11-20
> **Response to Reviewer 9VfQ**
>
> Thank you for your positive review! Below, we make one clarification and then respond to your weakness/question:
>
> 1. [*The critical observation is that the SOR iteration is bounded by a convex surrogate function with the parameter.*]
> While bounding the iteration cost of SOR by a surrogate function is indeed a key component of our approach, we note that the surrogate is not convex; instead the crucial properties of the surrogate are semi-Lipschitzness and unimodality.
> 2. [*Maybe a weakness of the paper is that the reduction to OCO is restricted to a certain type of algorithm (SOR) only so far. […] Is it possible to extend this framework to other algorithms for solving linear equations?*]
> It is indeed possible to extend the results to algorithms other than SOR; for example, in Sec. 2.5 we show similar guarantees for the widely used preconditioned conjugate gradient algorithm, although the parameter being tuned is still the relaxation parameter of the (Symmetric) SOR preconditioner. Future work could consider studying parameters other than the relaxation, such as the spectral bound of the Chebyshev semi-iterative method.

---

### Author Response · Authors · 2023-11-20
**General response**

We thank the reviewers for the positive reviews and helpful suggestions. In this response, we address two common points by (1) clarifying our view of the assumptions and (2) addressing concerns about limited experiments, providing new evaluations on linear system sequences generated by a numerical simulation. We have updated the draft with these results and some other fixes, and we would be happy to answer any further questions.

# On assumptions
Several reviewers correctly note that our theory applies either to a surrogate loss or makes a distributional assumption on the target vector. For the former, we only note that we do show empirically that the surrogate is fairly close in magnitude and shape to the quantity of interest (the iteration count) in Fig. 1 (left).

On the assumption about the targets ${\bf b}$: while we agree that it is restrictive in that it is unclear if many practical settings will have random targets, we view it to be reasonable “morally speaking” because we believe most scientific computing researchers would agree that—unless ${\bf b}$ is degenerate—it is the matrix ${\bf A}$ that has the dominant effect on the cost. Furthermore, we do give evidence that some regularity assumption on the targets might be theoretically necessary. Specifically, Fig. 2 (left) shows that we can force the cost to vary rapidly around an arbitrary optimal $\omega$ by setting the target to be a small eigenvector of the iteration matrix corresponding to that $\omega$ (in the figure we do it for $\omega=1.4$). While we have not shown anything formal, this is reminiscent of “needle-in-a-haystack” losses used to prove bandit lower bounds (Kleinberg et al., 2019).

# New experiments
Several reviewers note that the original experiments were rather simple, involving a sequence of i.i.d. linear systems. In our update, we add results for solving the 2D heat equation with a time-varying diffusion coefficient; while still a basic setup, such PDEs are commonly used to evaluate solvers. As a result, it likely better reflects what our approach would do in practice.

Note that the linear systems resulting from a standard time-stepping scheme here are positive-definite and have form ${\bf I}_n-\kappa_t{\bf A}={\bf b}_t$, where $\kappa_t$ is the time-varying diffusion coefficient and ${\bf A}$ is a fixed matrix. In our experiments we use this coefficient as context; in fact, multiplying by $c_t=1/\kappa_t$ shows that the linear system is mathematically equivalent to the diagonal-offset setting that we introduce in Sec. 2.4, where we show that contextual bandits can use knowledge of $c_t$ to approach instance-optimality (for simplicity we use $\kappa_t$, not $c_t$, as the context in experiments).

We evaluate how well our proposed methods set the parameter $\omega$ of SSOR-preconditioned CG; we choose this rather than just SOR because it is faster and more widely used in practice (in Sec. 2.5 we do show guarantees for tuning SSOR-preconditioned CG using Tsallis-INF). We report results on the right in Fig. 2 and in detail in App. G.2.

Fig. 2 (center-right, noting the log-scale) shows that across most matrix sizes ranging from around $10^3$ to around $10^5$, ChebCB and Tsallis-INF improve substantially over baselines such as vanilla (unpreconditioned) CG and SSOR-preconditioned CG with $\omega=1$ or $1.5$. Meanwhile, Fig. 2 (right) shows that ChebCB successfully recovers the instance-optimal trajectory, while Tsallis-INF converges to something close to the best fixed $\omega$.

While those results concern the iteration count, in Fig. 6 (right) we show that on larger problems this translates to runtime improvements of 2-3x over the $\omega=1$ baseline; for smaller problems solving is quick, so the cost is dominated by the overhead of learning. We view the harder, high-dimensional setting as more similar to those where learning will be applied in practice and note that the overhead does *not* scale with dimension; furthermore, it might be dramatically reduced by using a compiled language (we use MATLAB).

To put these numbers in perspective, note that Li et al. (2023) also propose a (deep) learned preconditioner for CG and test on a 2D heat simulation with tolerance 1E-8. Their preconditioner improves on SSOR with $\omega=1$ (Gauss-Seidel in their Tbl. 1) by 1.3x in runtime and 1.4x in iterations; on a comparably sized system ($n\ge10^4$), our ChebCB method obtains improvements of 2.5x and 3x, respectively. Of course, a direct comparison is hard without their code, which is not yet public (we have inquired about it); we provide more discussion and caveats in App. G.2.

Overall, we believe these new experimental results provide stronger evidence of the potential for accelerating numerical simulations in practice.

# References
Kleinberg, Slivkins, Upfal. *Bandits and experts in metric spaces*. J. ACM, 2019.
Li, Chen, Du, Matusik. *Learning preconditioners for conjugate gradient PDE solvers*. ICML 2023.

---

### Meta-Review · Area_Chair_SUri · 2023-12-05

**Metareview:**

The authors provide a framework to analyze optimal paramter setting for a specific class of iterative linear solvers, based on casting the problem in an online convex optimization framework and finding a low-regret algorithm. The reviewers were unanimous in their appreciation of the paper, and minor issues were well addressed during the rebuttal phase. Hence I recommend that the paper be accepted.

**Justification For Why Not Higher Score:**

While the paper provides a strong solution to a well-defined and well-motivated problem, it is unclear to me that this would be of broad interest to the ICLR audience due to the narrow problem scope.

**Justification For Why Not Lower Score:**

Theoretically and empirically justified results solving a well-motivated problem (solving a linear system using an iterative method).

---

### Decision · Program_Chairs · 2024-01-16

Accept (spotlight)